# Iridophores as a source of robustness in zebrafish stripes and variability in *Danio* patterns

Alexandria Volkening [1,2] & Björn Sandstede [1]

Zebrafish (*Danio rerio*) feature black and yellow stripes, while related *Danios* display different patterns. All these patterns form due to the interactions of pigment cells, which self-organize on the fish skin. Until recently, research focused on two cell types (melanophores and xanthophores), but newer work has uncovered the leading role of a third type, iridophores: by carefully orchestrated transitions in form, iridophores instruct the other cells, but little is known about what drives their form changes. Here we address this question from a mathematical perspective: we develop a model (based on known interactions between the original two cell types) that allows us to assess potential iridophore behavior. We identify a set of mechanisms governing iridophore form that is consistent across a range of empirical data. Our model also suggests that the complex cues iridophores receive may act as a key source of redundancy, enabling both robust patterning and variability within *Danio*.

[1] Division of Applied Mathematics, Brown University, 182 George Street, Providence, RI 02912, USA. [2] Present address: Mathematical Biosciences Institute, Ohio State University, 1735 Neil Avenue, Columbus, OH 43210, USA. Correspondence and requests for materials should be addressed to A.V. (email: volkening.2@mbi.osu.edu)

Characterized by stripes across its body and fins, the zebrafish (*Danio rerio*) has emerged as the archetype for studying pattern formation in vertebrates[1–3]. Zebrafish are amenable to experimental analysis, and numerous results (e.g., refs.[4–6]) have shown that their namesake patterns emerge robustly because of the self-organizing interactions of pigment cells. Other members of the *Danio* genus display markedly different patterns, giving the study of cell interactions on zebrafish evolutionary and mathematical value[2,7,8]. Until recently, the biological community has focused on two cell types, but new work[2,9–11] has uncovered the leading role of a third, iridophores[12]. The purpose of this work is to contribute to a better understanding of these newly uncovered dynamics from a mathematical modeling perspective. Through an agent-based approach that works alongside the empirical literature[11], we conduct mutational analysis in silico to help elucidate cell behavior. Our results suggest that iridophores are more than the leaders of stripe formation on zebrafish: through built-in redundancy in the cues they receive from other cells, iridophores may also serve as a source of robustness and variability within *Danio*.

As a zebrafish develops from a larva to an adult, 4–5 dark stripes and 4 light interstripes, represented by a layered mosaic of 3 main types of cells, emerge sequentially[2], Fig. 1. While melanophores are restricted to stripes, xanthophores and iridophores are spread across the skin, appearing in a loose (yellow or blue, respectively) form in stripes and adopting a dense (orange or silver, respectively) form in interstripes[6,13–15]. Thus stripes consist of yellow xanthophores atop blue iridophores and a bottom layer of black melanophores; interstripes, in turn, are made up of orange xanthophores above silver iridophores[11,13]. Until recently, empirical work (e.g., refs.[4–6,16]) focused on uncovering how melanophores and dense xanthophores interact through birth, competition, and movement. Iridophores were considered largely unnecessary, and a series of interactions in the form of short-range activation and long-range inhibition[17,18] was deduced[19]. Past mathematical models, whether discrete[20–24] or continuum[4,19,20,25–27], have also explored the interactions of melanophores and dense xanthophores.

The empirical picture changed significantly in recent years: new experimental results[9,10,28] discovered that iridophores play a leading role in patterning. By changing form between loose and dense, iridophores instruct the behavior of melanophores and

xanthophores and drive the sequential appearance of body stripes[2,9]. The importance of this governing behavior is particularly noticeable in *shady*, a mutant that lacks iridophores and instead features spots[11]. Recent research[29,30] also hypothesizes that another spotted phenotype (*leopard*, encoding connexin 41.8) contains dense iridophores that fail to change into loose form normally, further highlighting the critical dependence of stripe development on iridophore form.

While it is clear that carefully orchestrated transitions in iridophore form are necessary for stripe patterns to develop[2,9], what drives this dynamic remains largely unknown and speculative[3,7,12,29]. The mechanisms at work are complex and possibly nonlinear[3,29] but studying the signals iridophores receive from other cells has the potential to point to new communication pathway proteins, some of which may have orthologs in humans[30–32]. This is a place where mathematical methods can contribute to current biological unknowns, and this study is a first modeling step toward a better understanding of the interactions that govern iridophore form. Because several recent studies[10,28,31] have begun to uncover how iridophores signal other cells, we instead focus on helping elucidate the cues iridophores receive.

Here we develop a mathematical model that incorporates known interactions between xanthophores and melanophores and allows us to propose new potential mechanisms governing iridophore form transitions. In particular, using a mutational analysis approach that imitates the empirical work[11] in silico, we evaluate a range of candidate iridophore interactions by requiring agreement with pattern formation on wild-type zebrafish and mutants lacking cell types. Our work identifies a single set of nonlinear cues for iridophores that is consistent with these patterns. Further testing this model in silico, we demonstrate agreement across additional empirical observations, including the *choker* and *puma* mutants[11,33,34], cell ablation[4,10,35], and quantitative measurements[6,31,33]. In addition to providing a consistent answer to the question of iridophore form transitions that can be empirically tested, our model proposes that these cells are at the heart of robust patterning on zebrafish. Moreover, removing specific form transition signals between xanthophores and iridophores produces patterns present on other members of the *Danio* genus, suggesting a defining feature of zebrafish and offering a place where mutations may have critically impacted cell behavior to lead to variability within *Danio*.

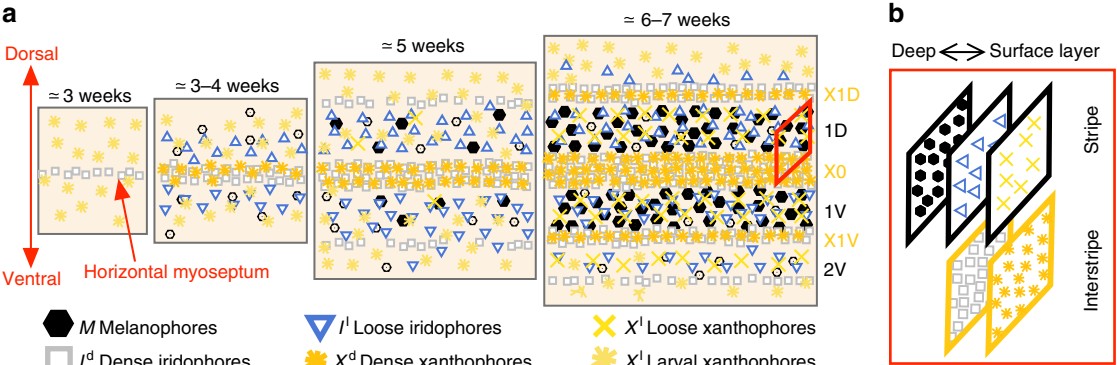

**Fig. 1** Cartoon of wild-type patterns on the body of zebrafish. **a** Patterns form sequentially outward from the central interstripe, labeled X0, with additional dorsal stripes and interstripes labeled 1D, X1D, 2D, X2D from the center (horizontal myoseptum) dorsally outward (similarly, ventral stripes and interstripes are labeled 1V, X1V, 2V, etc); see Supplementary Fig. 2 for zebrafish anatomy. Iridophores spread outward from the center (at the horizontal myoseptum), transitioning into loose to mark stripes and aggregating in dense form to produce new interstripes[2,9]. Xanthophores, initially spread across the body in loose or larval form, respond to dense iridophores by adopting a dense form themselves[10,15,31]. Melanophore birth relies on long-range cues from xanthophores and iridophores[10,19]. Timeline is redrawn from schematic by Singh & Nüsslein-Volhard[2] with different symbols used. **b** Cells are arranged in one-cell thick layers, with xanthophores at the surface, iridophores in the middle, and melanophores (present only in stripes) in the deepest layer (a fourth layer of so-called L-iridophores forms after patterns have developed)[6,13–15]

## Results

**Model overview and framework.** We begin by giving a general overview of our model, which is based on known interactions between melanophores and xanthophores and next turn to addressing open questions surrounding iridophore form transitions. To address the challenges stemming from these unknowns in a biologically faithful way, we break the available data into sets: a "derivation set" is used to derive iridophore interactions, and an "evaluation set" is reserved for evaluating the final proposed behaviors (Fig. 2). Notably, the evaluation set does not lead to further refinement of our iridophore mechanisms. Our main contribution is the identification of a network of cellular cues driving iridophore form changes that is consistent across this spectrum of experimental data. We conclude with a study of pattern robustness and suggest implications for *Danio* evolution.

We model pigment cells as individual agents (point masses) of five types: melanophores ($M$), loose and dense xanthophores ($X^l$, $X^d$), and loose and dense iridophores ($I^l$, $I^d$), Fig. 3a. Broadly, cells self-organize through birth, death, migration, and loose/dense form transitions in growing two-dimensional (2D) domains. We assume uniform epithelial growth[7], so cell positions move with the domain as it grows. To simplify our model, we account for the three-layer fish skin indirectly by constraining cell birth (to prevent overcrowding) only by cells that would appear in the same layer (see Fig. 1 and note that $M$ birth is an exception). Our initial condition approximates zebrafish at 21 days postfertilization (dpf), when patterning begins[3,9,15], and we simulate development by day until juvenile stages ($\approx$75 dpf). (Note that we follow the empirical custom of reporting time in developmental stage, as well as in dpf; see Table 1 and Methods for more

details on our domain sizes and growth rates, approximated from refs.[11,36].).

Cells interact on the fish skin at both short (neighboring cells) and long (half a stripe width) range, possibly regulated by direct contact[35], diffusion of secreted factors[28], dendrites[37], or longer extensions (long pseudopodia-like projections[38] or airinemes[39]). To model a cell or precursor communicating with its neighbors,

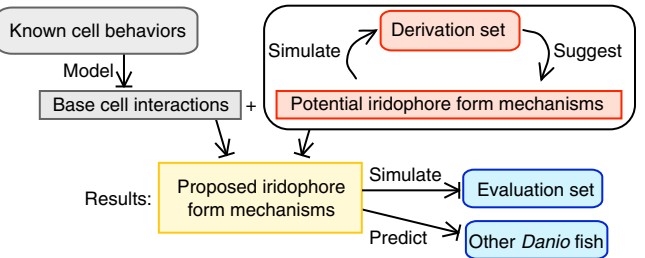

**Fig. 2** Overview of model development and results. See Supplementary Fig. 1 for details. Our model is built on known cell interactions and quantitative measurements and uses a "derivation set" (consisting of wild-type development and mutants lacking cell types) to identify potential (currently unknown) iridophore form transition mechanisms. By requiring consistency across the derivation set, we deduce a single network of iridophore interactions that can reproduce these patterns. This network is our main result; we then evaluate our model by simulating additional well-understood experiments ("evaluation set"). Because the model performs well, we use it to study additional unknown dynamics: in particular, we suggest how the patterns on two other *Danio* fish may have evolved from zebrafish

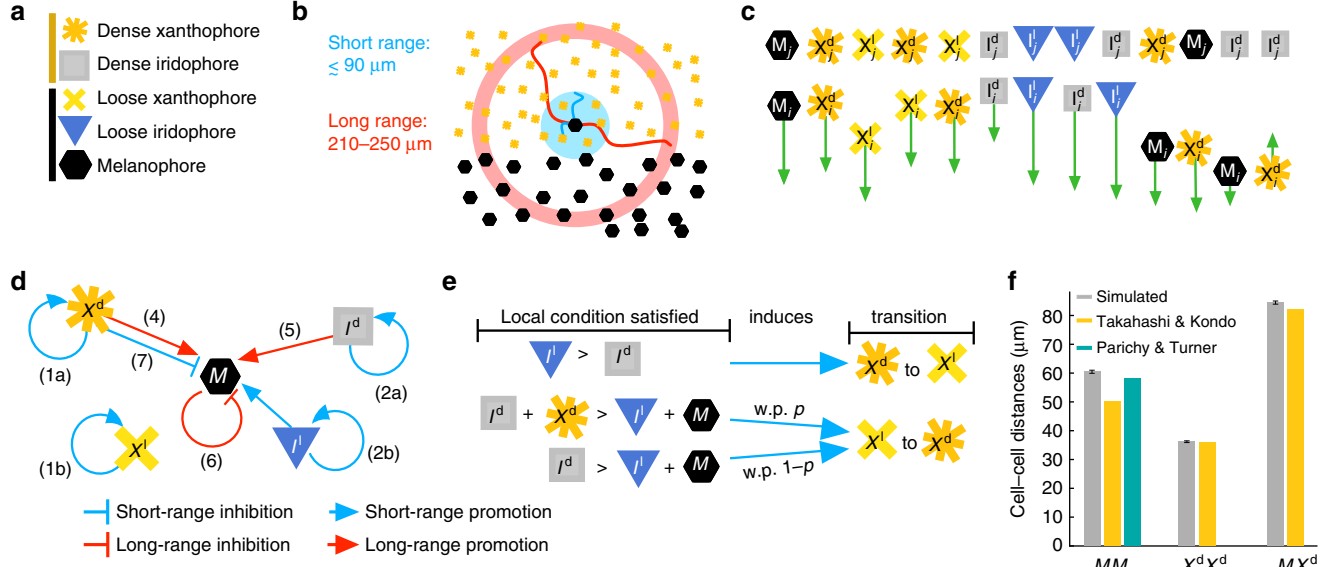

**Fig. 3** Model overview. See Methods for details. **a** Symbols used for cell agents. **b** Length scales are motivated by measurements of cell extensions[31,37-39]; interactions depend on the number of cells in local disk neighborhoods and a long-range annulus surrounding the point, or cell, of interest. Example shows an $M$ cell extending dendrites (short, blue) and long pseudopodia-like projections (red) toward $X^d$. **c** Migration, modeled by ODEs, is given by repulsive and attractive forces at short range[11,16,35,37,41-43]; arrow shows how $i$th cell moves in response to $j$th cell; length of arrow, while not to scale, indicates force strength (e.g., $M$ are repelled more strongly from xanthophores than from iridophores[11]); cell separation symbolizes that some cells are repelled or attracted over longer length scales than others (see Fig. 3f). **d** Birth and death rules largely dictated by[9,10,14,15,19,35,43]: xanthophores and iridophores are born through division of existing cells (1a–2b); $M$ appear in situ at random locations from precursors based on long-range promotion by $I^d$ and $X^d$ (4–5) and long-range inhibition by $M$ (6); and $X^d$ compete at short range with $M$ (7) but promote $M$ at long range (4). **e** Xanthophores receive signals to change form dependent on the proportion of cells locally (in a short-range disk around the xanthophore of interest); with probability $p$, $X^l$ react to $X^d$ (see Supplementary Note 2 for details). **f** Data for parameter fitting: average distance between neighboring cells at developmental stage J+ (see Table 1) compared to our approximations of data from Takahashi & Kondo[16] and Parichy & Turner[33]. Distances for mutants and other cells in Supplementary Fig. 10a; simulated data averaged across 100 simulations with error bars denoting standard deviation

## Table 1 Fish growth and pattern development

| Stage[a] | SSL[b] | Day (dpf) | Target pattern milestone[c] |
|---|---|---|---|
| PB | 7.5 | 21 | Faint strip of $I^d$ at center |
| PR | 8.5 | 30 | $I^l$ spreading across 1D & 1V |
| SP | 9.5 | 39 | $I^d$ present in X1V & X1D |
| SA | 10.1 | 44 | X1D & X1V forming |
| J | 10.9 | 51 | 2V & 2D faint |
| J+ | 13.0 | 70 | 2V & 2D forming |

Table provides a means of converting between measurements of zebrafish growth in our model. SSL and day provided at the start of stage
[a]Stages abbreviated: pelvic bud (PB), pelvic ray (PR), squamation onset posterior (SP), squamation through anterior (SA), and juvenile (J)[36]
[b]SSL[36] is a measurement of average fish length associated with standard length, which is measured in mm (see Supplementary Fig. 2). We simulate the full width of the fish body and roughly a third of its length (excluding a region around the eye)
[c]Days and pattern milestones (approximated from images in refs. [11, 36])

we consider disk and annulus neighborhoods centered at the point of interest. Disks, with radii on the order of the distance between two cells ($\approx$30–90 µm[16,31,33]), account for short-range interactions, and an annulus with an outer radius of 250 µm represents long-range dynamics (in comparison, stripes and interstripes on adult fish are approximately 600[38] and 400 µm wide[40], respectively), Fig. 3b. Our birth, death, and form transition rules depend on the proportion of cells in these neighborhoods, with some noise included where appropriate. For example, the $i$th melanophore at position $\mathbf{M}_i$ will die when the ratio of the numbers of $X^d$ to $M$ cells in a disk around $\mathbf{M}_i$ is too high. This is motivated by Nakamasu et al.[19], who showed that $X^d$ compete with $M$ locally. Our proportions reflect the stochastic nature of cell communication: enough $X^d$ must be present in a local disk around $\mathbf{M}_i$ for cell extensions to find their target at least once. Using disk and annulus neighborhoods to model cell communication avoids introducing directionality assumptions into our model.

Patterning evolves autonomously as the collective result of cell interactions. While the cues governing iridophore form are largely unknown, our birth, death, and migration rules are based closely on biological data and include some melanophore and xanthophore behaviors in a similar way as previous models[20,24]. During our model development, the behaviors involved in selecting xanthophore form also became better understood through experimental techniques[31]. We describe how we model these known cell behaviors, which we refer to as "base cell interactions", in detail in Methods (see Supplementary Methods and Supplementary Tables 1–4 for simulation conditions and parameters). Briefly, migration is modeled through ordinary differential equations (ODEs), which specify repulsive or attractive forces at short range, based on refs.[11,16,35,37,41–43]. All other behavior takes the form of discrete-time rules: each day, all cells on the domain are evaluated for possible death or form transitions (melanophores may die, but xanthophores and iridophores only change form). Cell birth occurs at randomly selected locations (from uniformly distributed precursors, in the case of melanophores[15,44,45], and existing cells, in the case of xanthophores and iridophores[14,15,35]). Our iridophore form rules, the main contribution of our model, are developed next.

**Deriving proposed iridophore form change mechanisms.** Iridophores guide the appearance of other cells on the fish skin by careful transitions in form between loose and dense, but what drives these form changes remains largely unknown[12,31]. It is clear that their behavior is complex: as shown in Fig. 1, stripe

development begins with the appearance of a strip of silver $I^d$ at the center of the fish, but at a delayed time, $I^d$ transition to loose and spread outward[2,9]. These spreading $I^l$ later localize at a distance, become dense, and induce $X^l$, which are initially randomly distributed across the fish skin, to become dense themselves[9,10]. Responding at long range, $M$ differentiate in situ from precursors to complete each stripe[9,10]. This process, driven by spreading and transitioning iridophores, continues dorsally and ventrally to produce additional stripes and interstripes.

One can envision various dynamics that could drive iridophores, and the space of possibilities has few limits. For example, it is unclear whether $I^d$ transition to loose concurrent with, because of, or to catalyze the birth of $M$. To untangle the complex dynamics present in wild type, we analyze the simplified patterns of mutants lacking cell types[46], Fig. 4c, d. These are mutated phenotypes that arise because some cell type fails to develop and consist of *pfeffer*[5,6,43] (no xanthophores; encodes csf1rA), *nacre* (no melanophores; encodes mitfa)[5,47], and *shady* (no iridophores; encodes ltk)[11,48]. Biologists[11] also consider double mutants of these fish, reducing the number of cells to one main type (e.g., *nacre;pfeffer* has only iridophores). Based on transplantation experiments[5,6,11], we assume these phenotypes emerge only because cells are missing; crucially, no interactions are altered.

Our approach to identifying form-change mechanisms for iridophores is built on this observation and imitates the mutational analysis experiments of Frohnhöfer et al.[11] in silico. We use *pfeffer*, *nacre*, and *pfeffer;nacre*, the three mutants that contain iridophores but lack one or more other cell types, to tease out insight into these cells and search for a single model that can simultaneously explain the development of these altered patterns and wild-type stripes (Fig. 2). For wild type (because there is more data available), we require matches at the major events in the timeline of pattern development (Table 1); in particular, we aim to produce simulations in which interstripes X1V and X1D begin to form at stage SP ($\approx$39–44 dpf), with $I^d$ appearing before $X^d$. Stripes 2V and 2D, in turn, should appear faint at stage J (beginning at $\approx$51 dpf) and be fully formed by stage J+ (beginning at $\approx$70 dpf; these points are based on our approximations of refs.[2,11,36]). Our second measurement of wild-type quality is concerned with bulk simulations: we quantify success by a low percentage of interruptions in interstripes across our stochastic simulations (stripes occasionally develop breaks on real fish). Note that we used *shady*, *pfeffer;shady*, and *nacre;shady*, the remaining three mutants, which do not contain iridophores, to specify xanthophore form rules (now more empirically understood) and to ensure that our iridophore interactions complement the rest of our model.

We now describe the mutants that contain iridophores but lack other cell types, namely *pfeffer*, *nacre*, and *pfeffer;nacre*[5,6,11,43], Fig. 4c. First, *pfeffer* is characterized by dark spots (aligned horizontally) and peppered $M$ in a silver sea of $I^d$[5,43]. Where sufficient $M$ are present locally, iridophores appear loose[11]. *Nacre*, in turn, features an expanded central interstripe with rough borders flanked ventrally by patches of $I^d$ and $X^d$ against a blue $I^l$ background[11] (note this can be seen more clearly in Supplementary Fig. 6). Frohnhöfer et al.[11] consider the blue spots of *pfeffer* and orange patches of *nacre* to be remnants of wild-type stripes. In contrast, *pfeffer;nacre* has a silver pattern consisting of $I^d$[11].

*Pfeffer;nacre* demonstrates that iridophores do not control their own transition from dense to loose; based on *pfeffer* and *nacre*, however, we know melanophores and xanthophores each provide partial cues. If this were not the case and, for example, only melanophores were involved, the patterns on *nacre* and *pfeffer;nacre* would be identical. Because $X^l$ initially cover the fish skin in the locations of both future stripes and interstripes[15], we assume these cells have a largely passive role; they adopt a dense form

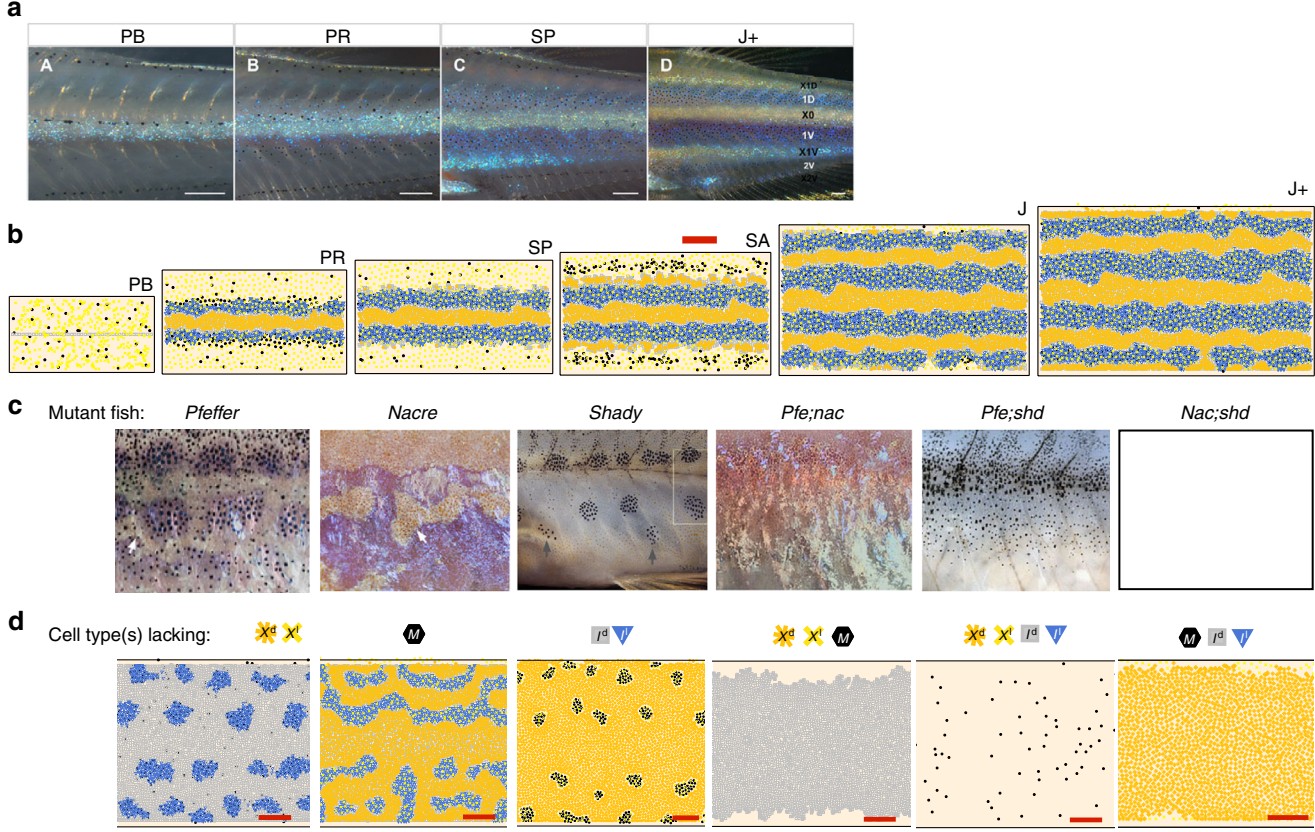

**Fig. 4** Representative simulations of the proposed model on derivation set. Scale bar is 500 μm for simulated images. **a** Wild-type development[11] (scale bar = 250 μm) and **b** model simulation (domain size accounts for roughly a third of the fish body length and its full height); also compare with Fig. 1. **c** Mutants (and double mutants) lacking cell types and **d** simulations (shown at J+; note images of *nacre;shady* are not empirically available; see Supplementary Note 2 and Supplementary Fig. 17 for a discussion of how this shaped our xanthophore dynamics). Animations of daily development from 21 dpf are given in Supplementary Movies 1–5. Experimental images (**a**, **c**) reproduced from Frohnhöfer et al.[11] and licensed under CC-BY 3.0 (http://creativecommons.org/licenses/by/3.0); published by The Company of Biologists Ltd

after $I^d$ appear[10,31] but do not lead the dynamics. Thus our model for iridophore form will consist of cues from $M$ and $X^d$. Mahalwar et al.[31] have also suggested that signals from both melanophores and xanthophores may be needed to induce iridophore transitions from dense to loose.

The simplest explanation for the dark spots in *pfeffer* is that: [A] $M$ transmit a local signal to $I^d$ to become loose, Fig. 5a. When $M$ are not present, iridophores could revert back to dense form, leading to the reciprocal mechanism: $[\tilde{A}]$ When $I^l$ lose a local signal from $M$, they become dense. Because $I^d$ transition to loose to complete the central interstripe X0 in *nacre*, we suggest $I^d$ respond to a long-range signal from interstripe cells. This signal could come from $X^d$ or $I^d$, but based on *pfeffer;nacre*, $X^d$ are responsible. Notably, Patterson et al. have suggested that an abundance of xanthophores can repress the appearance of dense iridophores locally[28], and Frohnhöfer et al. have hypothesized that xanthophores are needed to halt expanding interstripes, providing a long-range inhibitory effect on $I^d$[11]. Thus the next mechanism is: [B] When $I^d$ sense sufficiently many $X^d$ at long range, they become loose. The counter mechanism explains the presence of $I^d$ patches in *nacre*: $[\tilde{B}]$ When $I^l$ lose a long-range signal provided by $X^d$, they adopt a dense form. Lastly, the close association of $X^d$ with $I^d$[11,15,31] motivates a local signal from $X^d$: [C] $X^d$ maintain $I^d$ in dense form, and $[\tilde{C}]$ $X^d$ induce $I^l$ to become dense. Note that [A], [B], and [C] represent cues melanophores and xanthophores may transmit to $I^d$, while $[\tilde{A}]$, $[\tilde{B}]$, and $[\tilde{C}]$ refer to signals transmitted to $I^l$.

Based on *pfeffer* and *nacre*, we know both melanophores and xanthophores are involved in specifying iridophore form, so mechanisms [A], [B], and [C] and $[\tilde{A}]$, $[\tilde{B}]$, and $[\tilde{C}]$ (Fig. 5a), which involve, respectively, only $M$, long-range $X^d$, or short-range $X^d$, need to be combined. Therefore, we considered combinations of these building blocks with the goal of capturing the patterns in our derivation set. Our main finding is that the interactions [A||(B&C)] and $[\tilde{A}\&(\tilde{B}||\tilde{C})]$ are the minimal rules we identified consistent with these patterns; see Fig. 5b, c. The first rule says that dense iridophores respond independently to signals from $M$ and $X^d$ to become loose: $M$ locally are enough to induce form transitions; alternatively, $I^d$ will also become loose when $X^d$ density is sufficiently low locally but high at long range. According to the latter rule, the cues $I^l$ require to become dense combine requirements on both $M$ and $X^d$: $M$ density must be low locally and, additionally, if $X^d$ density is either high locally or low at long range, a form transition occurs. We recognize these rules are complex and nonlinear (as has been hypothesized in refs.[3,29]): where & appears in our rules, we require dual conditions to be met simultaneously, suggesting that nonlinear signaling is present. On the other hand, || signifies that more than one independent set of cues is capable of selecting iridophore form; see Methods. As discussed next, relaxing the requirement that these conditions are met simultaneously or removing mechanisms causes the model to lose consistency with one or more mutants lacking cell types, and we interpret this complexity as a source of pattern robustness and variability within *Danio*.

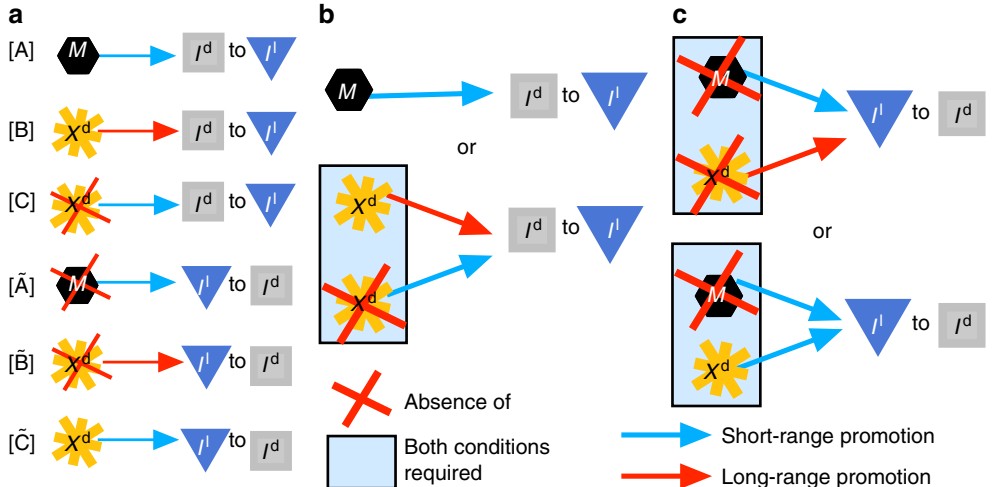

**Fig. 5** Iridophore form dynamics. **a** Building block mechanisms inferred from *pfeffer* and *nacre*. Each rule prescribes the condition that must be met at short (blue) or long (red) range in order for an iridophore form transition to occur. For example, [B] states that the presence of $X^d$ at long-range causes $I^d$ to become loose. **b**, **c** Our final model for governing iridophore form is obtained by considering combinations of these building blocks in search of consistent agreement across our derivation set (wild-type and 6 mutants lacking cell types); note that [A], [B], and [C̃] may be active signals, while [C], [Ã], and [B̃] could be interpreted as passive cues that represent the absence of a signal from the crossed-out cell type. **b** In particular, we propose $I^d$ become loose if either of the two conditions is fulfilled: if there are sufficient $M$ locally, a dense-to-loose transition occurs; alternatively, if $X^d$ are absent locally, but present at long range, a dense-to-loose transition is also induced. **c** Loose-to-dense transitions do not occur unless $M$ are absent locally, but absence of $M$ is not sufficient to induce a transition; additionally, one of the two conditions on $X^d$ must be met: either $X^d$ are absent at long range, or $X^d$ are present locally. Note that, from a modeling perspective, the red X means "too few" cells of the specified type not necessarily complete absence (biologically, "too few" may correspond to "absence" if there are not enough cells present for communication to occur in the stochastic environment on the fish skin)

**Overview of model simulations.** Here we present representative results drawn from stochastic simulations of our proposed model: we first demonstrate consistency with the derivation set (Fig. 4), then present model performance on our evaluation set (Fig. 6 and Supplementary Figs. 12–15) and conclude by showing how simplifications in the proposed iridophore mechanisms may be related to robust pattern formation and *Danio* evolution (Figs. 7 and 8). Regardless of whether we show a timeline of development or only the final image, every image was simulated from an initial condition that approximates the fish at 21 dpf, when adult patterning begins with the appearance of a strip of $I^d$ at the center of the fish[2]. All simulations use domain sizes that roughly track the full width of an average zebrafish and a third of its patterned body length.

**Consistency of iridophore interactions with derivation set.** Because the cues that determine iridophore form are largely unknown empirically[7,12,29], we address the wide space of possibilities by requiring that our interactions are consistent across wild type and mutants/double mutants lacking cell types. Model performance on these patterns, our derivation set, is shown in Fig. 4 in relation to experimental images. In agreement with the biology, wild-type pattern formation (Fig. 4a, b, Supplementary Fig. 4, and Supplementary Movies 1 and 2) progresses autonomously from the point at which $I^d$ appear in the center of the fish. We meet the target pattern milestones in Table 1, and our simulations have a success rate of producing unbroken interstripes of 89% across 100 simulations. Simulated mutants lacking cell types are collected in Fig. 4c, d in comparison to empirical images (also see Supplementary Figs. 5–9 and Supplementary Movies 3–5). We stress that all simulations rely on the same cell interaction network; the only change made to simulate *pfeffer*, for example, is to turn off $X^l$ and $X^d$ birth. Measurements of cell speed, stripe width, and cell-to-cell distances that demonstrate further consistency with experimental data are given in Supplementary Figs. 10 and 11.

**Performance of iridophore interactions on evaluation set.** Once we identified a model that could account for wild type and mutants lacking cell types with parameters calibrated to quantitative measurements, we simulated additional experiments reserved to evaluate the model. These are patterns that develop on zebrafish due to some known adjustment (e.g., ablation may disturb the initial pattern[4,10,35]). We make the adjustment dictated by the biological literature and test whether the patterns produced by our model agree; because this portion of the modeling process is retrodictive rather than predictive, we include a few examples in Fig. 6 and refer to Supplementary Figs. 12–15 for additional examples (for detailed simulation conditions, see Supplementary Methods). We find our model performs well across the evaluation set, which includes ablation[4,10,35], *choker*[11] and *puma*[33] mutants, and quantitative measurements[6,31,33].

Briefly, Yamaguchi et al.[4] showed that laser ablation of a rectangular region of melanophores and xanthophores resulted in patterns with lost directionality but maintained width, Fig. 6a. Our simulated ablation (Fig. 6b) also disrupts alignment while upholding stripe width. *Puma* (encoding bbc3), in turn, features a severe reduction in $M$ birth[34]; its phenotype (Fig. 6c) and our simulated version, obtained by reducing $M$ birth without any other model changes (Fig. 6d), feature spotty broken stripes. *Choker*, encoding meox1[2] (Fig. 6e), lacks the central strip of $I^d$ present at 21 dpf (see Fig. 1a); instead, these cells appear randomly at stage PR[11]. In good agreement with Frohnhöfer et al., who showed the initial iridophores at the center of the fish help align stripes[11], our simulated *choker* mutant (Fig. 6f) has an undirected, labyrinth-style pattern. Lastly, as an example quantitative measurement, Fig. 6g shows the evolution of the average coefficient of variation (CV) for the distance between neighboring $M$ across 100 wild-type and 80 *pfeffer* simulations. CV is an empirical measurement of pattern quality defined as $CV = 100 \times$ standard deviation/mean, and should decay in time, symbolizing better formed patterns[6,33].

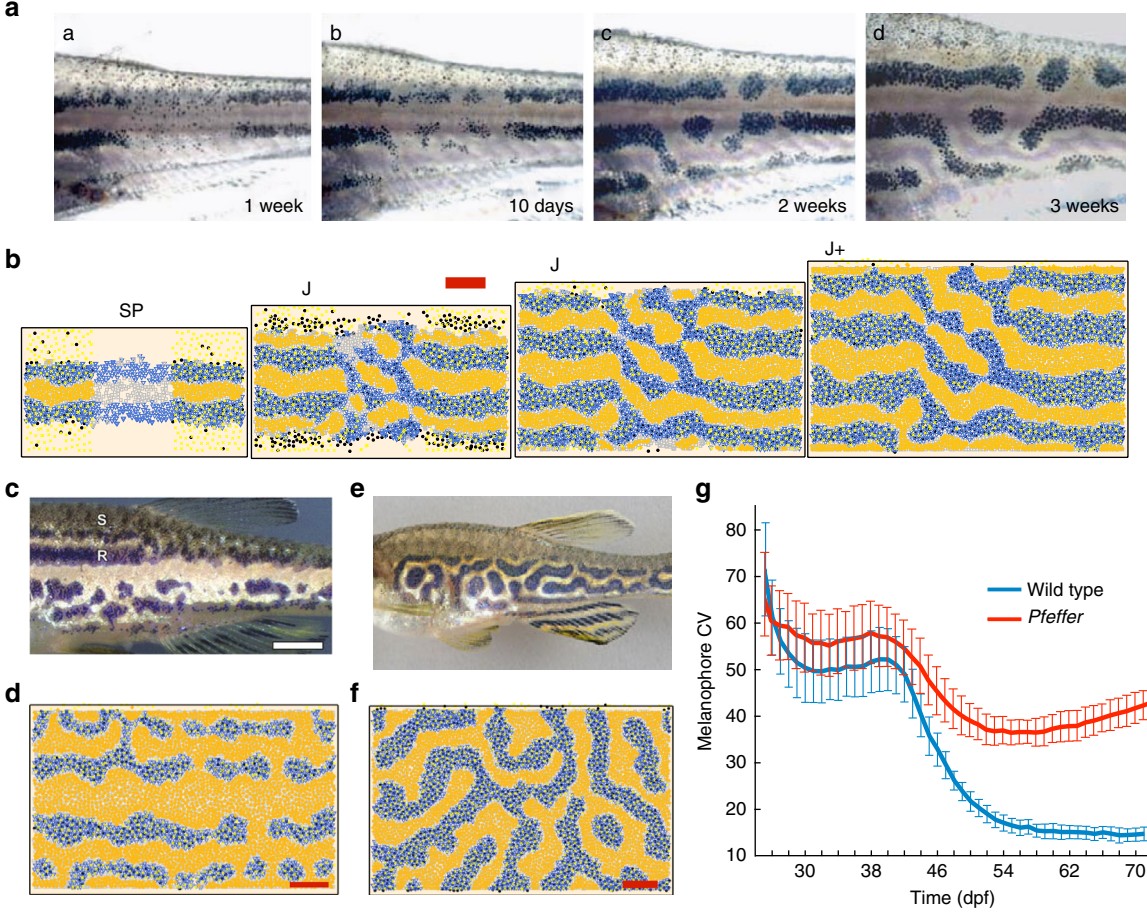

**Fig. 6** Model performance on sample dynamics from evaluation set. This is a set of well-understood dynamics reserved for model evaluation. All simulation scale bars are 500 μm. **a** Ablation of a rectangular region in vivo[4] and **b** ablation in silico (also see Supplementary Movie 7). In both cases, the resulting patterns are characterized by stripes with normal width but lost directionality. Ablation is simulated at 40 dpf (when our pattern seems to most closely resemble the first image in the timeline (**a**)) by removing all melanophores and xanthophores and 20% (randomly selected) of loose and dense iridophores (motivated by the indirect effects of laser ablation on iridophores described in ref. [7]) in a rectangular region 1000 μm long (and the full height of the domain). **c** *Puma* involves a strong reduction of *M* birth (scale bar is 2 mm)[34]; **d** *puma* is simulated by reducing *M* birth by ≈85%. **e** *Choker* lacks the typical central strip of $I^d$ at 21 dpf; instead, these cells arise randomly at a later time[11]. **f** *Choker* is simulated by removing the $I^d$ strip from our initial condition and introducing 15 $I^d$ cells at random locations at 31 dpf (see Supplementary Movie 6). **g** *M* CV averaged across 100 wild-type simulations and 80 *pfeffer* simulations (error bars denote standard deviation) plotted versus time; empirical measurements of CV range from 100 to 30 during wild-type development based on our approximation of data in ref. [33], and higher CV corresponds to reduced xanthophores[6] (as shown, we find higher CV values in *pfeffer*, which lacks xanthophores). Image **a** reproduced from Yamaguchi et al.[4] with permission; Copyright (2007) National Academy of Sciences, U.S.A. Image **c** reproduced from Parichy et al.[34] with permission from Elsevier; Copyright (2003) Elsevier Science, U.S.A. Image **e** reproduced with minor adaptation from Frohnhöfer et al.[11] and licensed under CC-BY 3.0 (http://creativecommons.org/licenses/by/3.0); published by The Company of Biologists Ltd

**Implications for robust pattern formation and *Danio* evolution.** Simulating simplifications of our model serves as an additional test of the proposed iridophore dynamics in two ways: first, because cell communication is stochastic and imperfect on the fish skin, the model should be robust to changes. At the same time, if it is capturing biological truth, the model should also be able to account for features of other *Danio* fish when adjusted further, as it has been suggested that zebrafish stripes evolved from other *Danios* through gain-of-function mutations[8]. To evaluate the model on these two criteria, as well as explore the importance of the different components of our compound iridophore form mechanisms, we simulate reduced versions of the interaction network in Fig. 5.

Regarding robust patterning, Fig. 7a–c collect the wild-type patterns that emerge when dense iridophores fail to respond to (or receive) signals from $X^d$ and $M$. First, in Fig. 7a, we remove all signaling $X^d$ provide $I^d$ to become loose, so that only $M$ instruct $I^d$

to transition. In contrast, in Fig. 7b, we explore pattern formation when $I^d$ do not sense the presence of $M$. We can also test the importance of long-range signals in our model by removing the signal $I^d$ receive to become loose when $X^d$ density is high at a distance (Fig. 7c); biologically, this could mean $I^d$ lose their sensitivity to $X^d$ at long range or simply fail to contact any of these cells through dendritic extensions. In Fig. 7d–f, we explore the failure of cues inducing $I^l$ to become dense, by removing any sensitivity to the absence of $M$, disrupting local signals from $X^d$, or making $I^l$ unable to correctly react to $X^d$ both locally and at long range.

Across the reduced interactions in Fig. 7a–f, we find that wild-type patterning is remarkably robust. While the patterns that develop are messier than those produced by our identified network, differ in the finer details, or admit more frequent breaks across our stochastic simulations, stripes are nevertheless selected and form autonomously. Notably, these stripes develop despite

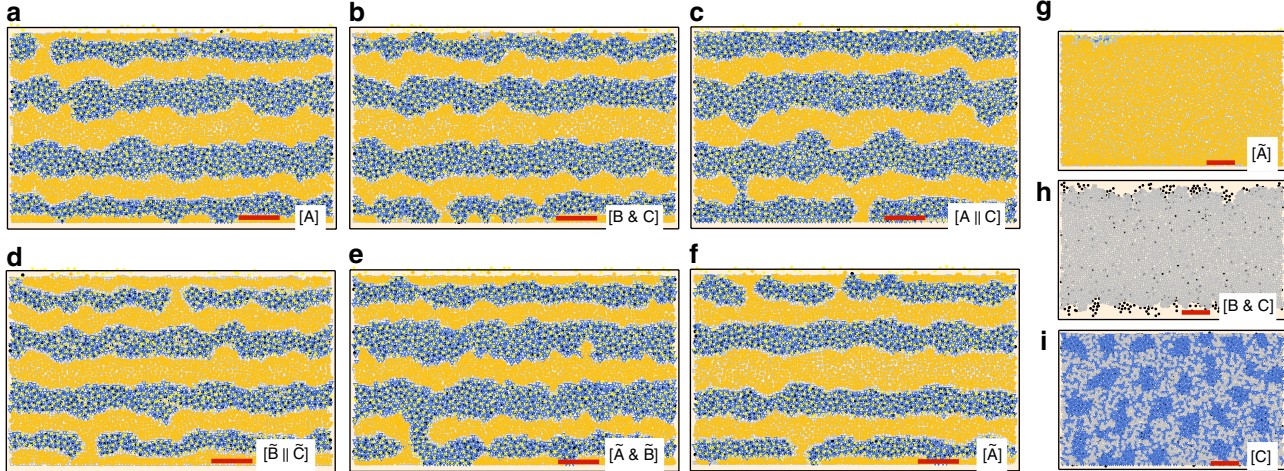

**Fig. 7** Robustness in pattern formation: dynamics of reduced model. Letter inset shows alternative mechanism used to control iridophore form; see Fig. 5. Wild-type patterns under changes in the signals $I^d$ receive to become loose: **a** complete loss of $X^d$ cues to $I^d$ to become loose; **b** loss of $M$ signals to become loose (e.g., $I^d$ fail to sense melanophores nearby); or **c** loss of long-range cues from $X^d$ for $I^d$ to become loose (e.g., $I^d$ lose any sensitivity to the absence of $X^d$ at long range). Wild-type patterns under reductions in the signals $I^l$ receive to become dense: **d** lack of sensitivity of $I^l$ to the presence of $M$ (e.g., $I^l$ fail to receive a signal from $M$ that supports their loose form); **e** disruption of local signals from $X^d$ (e.g., $I^l$ fail to receive a cue from $X^d$ to become dense); or **f** complete failure of correct reactions to $X^d$ both locally and at long range. In contrast to wild-type patterning, **g** nacre and **h**, **i** pfeffer patterns are not robust to loss of signals to iridophores. This offers an explanation for empirical observations[11] that mutants lacking cell types are more variable than wild-type stripes. Scale bar is 500 μm. See Supplementary Methods for details on the parameter values altered to create these images

complete failure of the specified cell interactions; on the fish skin, we expect that partial failure due to stochastic fluctuations is much more likely than complete loss. For example, we doubt that all $I^d$ cells fail to respond to signals from $M$ but rather that the noisy biological environment means occasionally some $I^d$ cells do not make contact with any $M$, essentially leading to a failure (lack) of signaling at those times. While our model suggests wild-type pattern formation is robust to altered interactions, mutants lacking cell types (Fig. 7g–i) are not. Our simulated mutants differ widely when cues are lost, leading to uniform patterns on nacre and pfeffer or other dynamics. Notably, Frohnhöfer et al. observed that mutants lacking cell types are more variable than wild-type stripes[11]. Thus the robustness of wild-type patterns and comparative fragility of mutant phenotypes produced by our model agree with empirical results[11].

Zebrafish (D. rerio) stripes are thought to have evolved from other Danio fish patterns by gain-of-function mutations (or other fish evolved from zebrafish through loss-of-function mutations); this means that some crosses of zebrafish with other Danio fish result in zebrafish-like stripes[8] (note that such early genetic experiments can only detect average effects across alleles). Our model is consistent with this hypothesis: as shown in Fig. 8a–f, our model suggests that the patterns on D. albolineatus and D. margaritatus could be related to zebrafish through a loss of iridophore form signaling. Interestingly, both altered patterns arise when we reduce the communication between xanthophores and iridophores. Under wild-type conditions, our model suggests the absence of $X^d$ at long range helps induce $I^l$ to become dense (and thus initiate the formation of new interstripes). The pattern observed on D. albolineatus forms when we remove this mechanism: if $I^l$ do not respond to the absence of $X^d$ at long range but maintain a sensitivity to $X^d$ locally, no new interstripes form (interestingly, Patterson et al.[28] found D. albolineatus-like patterns develop on zebrafish when a factor called Csf1, which is related to xanthophore–iridophore dynamics, is overexpressed). The reduced interactions consistent with D. margaritatus, on the other hand, remove a local signal from xanthophores to iridophores; we see spots because $I^d$, sensing $X^d$ at a distance

but not concerned with their local environment, break inter-stripes perpendicularly in the same way they halt the parallel advancement of light regions in wild type. This supports recent results[10,15,31] highlighting the close coupling of xanthophores and iridophores in zebrafish and suggests that it may be crucial signals between these two cells that distinguish zebrafish within its genus.

Notably, like our simulations of wild-type zebrafish, our patterns resembling D. albolineatus and D. margaritatus are robust themselves. As shown in Fig. 8, we find that two further model simplifications are consistent with each of these patterns. For example, D. margaritatus patterns form regardless of whether $M$ signals are impacted (Fig. 8e) or not (Fig. 8f). This could explain why these fish persisted from an evolutionary standpoint; our model suggests that, rather than being fragile mutants, D. albolineatus and D. margaritatus can withstand occasional failure of interactions on the fish skin. While we do not know of empirical data on mutations in these two Danios leading to missing cell types, we show what our model predicts D. albolineatus and D. margaritatus mutants lacking cell types should look like in Fig. 8g–k. Further studies of these fish will lead to a better understanding of where cell interactions diverged during Danio evolution and provide a place to test our results.

## Discussion

The framework of reaction–diffusion systems and Turing pattern formation[17] has been applied to modeling zebrafish stripes[4,19,20,26] and debated by the biological community[7,15,49,50]; in this setting, melanophores and dense xanthophores are often taken to be the diffusing substances. Partial differential equation models present overarching views of patterning in which many cell interactions are combined into a few parameters. In contrast, the mechanisms we identified for governing iridophore form are complex and nonlinear. They involve dual signaling from both melanophores and dense xanthophores and include interactions at short and long range. This was challenging initially: our mathematical intuition is to search for a single elegant rule. From a biological perspective, however, resilience to mutation is key,

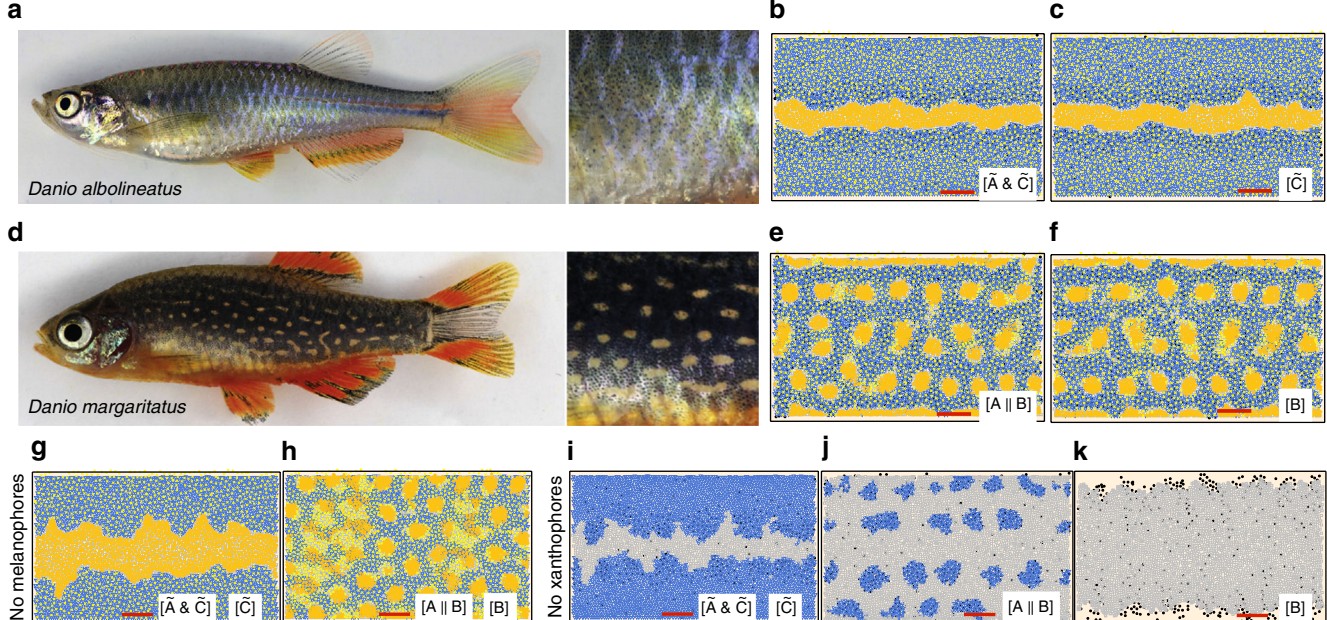

**Fig. 8** Variability in pattern formation. Dynamics of our simplified model with certain iridophore pathways removed are consistent with other *Danio* fish. **a** *Danio albolineatus* features a nearly uniform pattern, with a central light stripe near the tailfin. **b** Our model suggests *D. albolineatus* patterns differ from zebrafish stripes in a critical loss of long-range promotion of $I^d$ by $X^d$: in particular, the absence of $X^d$ at long range fails to induce $I^l$ to become dense (in other words, we obtain *D. albolineatus* patterns when we simplify our loose-to-dense iridophore transition rule from $[\tilde{A}\&(\tilde{B}||\tilde{C})]$ to $[\tilde{A}\&\tilde{C}]$, see Supplementary Fig. 3a). Note we assume *D. albolineatus* contains both $X^d$ and $X^l$; if only one type of xanthophore is present (e.g., refs. [28,39] suggest that the xanthophores on this fish have some characteristics in common with $X^d$), we predict *D. albolineatus* pattern evolution involved additional changes to xanthophore form behavior. **d** *D. margaritatus* is characterized by light spots; **e** our model indicates the $I^l$ on this fish are not sensitive to local support signals from $X^d$ that are crucial in zebrafish (this means we simplify our rule for dense-to-loose transitions from $[A||(B\&C)]$ to $[A||B]$; see Supplementary Fig. 3b). Notably, **c**, **f** may suggest why these altered patterns evolved as part of robust *Danio* species, rather than through fragile mutations: when we further reduce the iridophore pathways in our model by removing $M$ signaling to iridophores (by using mechanism $[\tilde{C}]$ in place of $[\tilde{A}\&\tilde{C}]$ or $[B]$ in place of $[A||B]$, respectively), we obtain the same patterns, suggesting that these fish are robust themselves. We do not know of mutant phenotypes lacking cell types for these *Danio* fish but provide what our model predicts *D. albolineatus* and *D. margaritatus* **g, h** without melanophores or **i-k** without xanthophores, respectively, would look like as a means of future model testing. See Supplementary Methods for the parameter values altered to create these simulations. Simulation scale bars are 500 μm. Empirical images **a**, **d** reproduced from Singh and Nüsslein-Volhard[2] with permission from Elsevier; Copyright (2015) Elsevier Ltd

and the fish skin environment in which cells self-organize is necessarily irregular and imperfect. One of the strengths of the Turing approach is that tuning parameters produces a diversity of patterns, but these models suffer from robustness issues: small parameter changes can lead to significantly different behavior[7]. Simulating simplified forms of our model offers an interpretation for its complexity in terms of pattern robustness and variability.

We suggest the richness of patterning by self-organization on zebrafish stems from the presence of redundancy in iridophore interactions. This redundancy can admit both failure of interactions without destroying stripes and simplifications that generate a diverse array of patterns. In particular, built-in backup in the cues iridophores receive from other cells can account for robust stripe formation: when one interaction governing iridophore form fails, the remaining cellular dynamics can, with the exception described next, still support wild-type patterning. The exception is that simplifying some interactions between dense xanthophores and iridophores critically breaks zebrafish, producing patterns consistent with relatives *D. albolineatus* and *D. margaritatus*. Related to empirical results[10,15,31] noting the close association between these two cell types, this indicates that local xanthophore–iridophore interactions may be a defining feature of zebrafish within *Danio*. In particular, because crosses of zebrafish with some other *Danios* produce zebrafish-style stripes[8], it has been suggested that zebrafish patterns evolved through gain-of-

function mutations: our model indicates that these gains may be in the form of additional, more complex signaling from xanthophores to iridophores.

Regarding future work, Mahalwar et al.[31] recently discovered xanthophores present in an intermediate form between loose and dense in some mutant fish, and some early larval and newly differentiating adult melanophores are known to interact with loose xanthophores through airineme extensions[39,51]. Accounting for these intermediate and larval cells in the next generation of models will allow for further testing. Additionally, this study focused on wild-type patterns and mutants lacking cell types: mutant phenotypes that arise from (often unknown) altered cell behavior, despite all cell types being present, will be the focus of our future work. From an analytical perspective, it would also be interesting to consider a continuum limit; it is possible, for example, that iridophore behavior could be accounted for with density-dependent motion[52]. A bifurcation analysis of the continuum limit, together with agent-based modeling, would provide a more comprehensive study of robustness and variability in zebrafish and other *Danio* patterns.

## Methods

**Model overview**. We consider independent cells interacting on growing 2D domains. Our cells are separated into five classes and modeled as point masses with

their positions given by their $(x, y)$ coordinates, with the following notation:

$$\mathbf{M}_i(t) = \text{position of } i\text{th melanophore at time } t;$$
$$\mathbf{X}_j^d(t) = \text{position of } j\text{th dense xanthophore at time } t;$$
$$\mathbf{X}_i^l(t) = \text{position of } i\text{th loose xanthophore at time } t;$$
$$\mathbf{I}_j^d(t) = \text{position of } j\text{th dense iridophore at time } t;$$
$$\mathbf{I}_i^l(t) = \text{position of } i\text{th loose iridophore at time } t;$$
$$N_M(t) = \text{number of melanophores at time } t;$$
$$N_X^d(t) = \text{number of dense xanthophores at time } t;$$
$$N_X^l(t) = \text{number of loose xanthophores at time } t;$$
$$N_I^d(t) = \text{number of dense iridophores at time } t;$$
$$N_I^l(t) = \text{number of loose iridophores at time } t.$$

We do not consider so-called L-iridophores, as these cells appear on the skin only after patterns have formed (they may be involved in later maintenance[11]). Additionally, we combine larval xanthophores and loose xanthophores into one class of xanthophores. We use periodic boundary conditions lengthwise for cell movement, birth, death, and form changes. Our boundary conditions at the top and bottom of the domain are no flux: we implement this by removing the vertical component of a cell's velocity if the cell is within 50 μm of the top (respectively, bottom) edge of the domain and the $y$ coordinate of its velocity points upward (respectively, downward). One full simulation time step, corresponding to 1 day of development, consists of (1) updating domain sizes for fish growth; (2) adjusting cell positions to model epithelial growth; (3) updating cell positions due to migration (10 iterations of forward Euler method with step size $h = 0.1$ day); and (4) simultaneously updating the domain for cell birth, death, and changes in form. Each of these model components will be described in detail below. For parameter values, initial conditions, and experiment-specific adjustments, see Supplementary Methods, Supplementary Tables 1–4, and Supplementary Fig. 18. Measurements of cell-to-cell distances, cell speeds, and stripe width are used to set parameters for base cell interactions and evaluate model output (Fig. 3f and Supplementary Figs. 10 and 11).

**Domain growth and development staging**. Because zebrafish growth rates differ depending on lab conditions, experimentalists often track development by reporting developmental stages (see Table 1) and associated standardized standard length (SSL)[1,36]. SSL is a measurement of average body standard length (standard length (SL) is measured from the tip of the fish snout to the proximal edge of the tailfin, see Supplementary Fig. 2) based on the development of reference zebrafish[1,36]. During the timeframe we are interested in, zebrafish grow in size from ≈7 to 16 mm SL[36], and their width, measured roughly at the beginning of the anal fin, increases from ≈1 to 3 mm (based on our approximations of data in ref.[36]). Our simulations, which begin at ≈21 dpf, track patterning on the full width of the fish and roughly a third of its standard length (after removing a 1.5 mm region for the eye), beginning at stage PB (onset at 7.2–7.5 SSL)[36]. Our initial domain is thus 1 mm high and 2 mm long. Each day cell positions are stretched across the domain in the vertical and horizontal directions to model their response to epithelial growth (this is done by multiplying a cell's $x$ and $y$ coordinates by respective scaling factors). We specify that domains expand by 38 μm per day length-wise and 27 μm per day in height. Note that Parichy et al. provide graphs relating standard length to dpf for some example fish in ref.[36]. SSL is also associated with developmental stage in ref.[36]. We rely on approximations of this data[36] and additional time-lapse images of development[11] to inform our daily growth rates and domain sizes.

**Cell migration**. Motivated by refs.[11,16,35,37,41–43], we specify repulsive and attractive forces between cells. Our ODEs are first order, so we consider a kinematic model in which velocity is directly proportional to force, a reasonable simplification since we are interested in the movement of small agents. All cells experience migration, but each cell type does not interact with all five types of cells: interactions are only included when necessary for model results and suggested by the biology (for example, there are no repulsive forces between $M$ and $I^l$ because these cells occupy the same region but in different layers on the fish skin). In all cases, the force a cell of type $\mu$ at position $\mathbf{A}$ exerts on a cell of type $\nu$ at position $\mathbf{B}$ is given by:

$$\mathbf{F}^{\mu\nu}(\mathbf{A}, \mathbf{B}) = -R^{\mu\nu}\left(\tfrac{1}{2} + \tfrac{1}{2}\tanh\left(\tfrac{r_{\mu\nu} - \|\mathbf{A} - \mathbf{B}\|}{\delta}\right)\right)\frac{\mathbf{A} - \mathbf{B}}{\|\mathbf{A} - \mathbf{B}\|}$$
$$= f^{\mu\nu}(\|\mathbf{A} - \mathbf{B}\|)\frac{\mathbf{A} - \mathbf{B}}{\|\mathbf{A} - \mathbf{B}\|}.$$

Here $^{\mu\nu}$ denotes that different parameters apply for interactions between different types of cells, and $\delta = 5$ smooths out the hyperbolic tangent (see Supplementary Fig. 18). We chose hyperbolic tangents because forces operate over a short scale, mediated by direct contact or dendrites.

The movement of the $i$th melanophore at position $\mathbf{M}_i$ is given by:

$$\frac{d\mathbf{M}_i}{dt} = \underbrace{\sum_{j=1, j\neq i}^{N_M} f^{MM}\left(\|\mathbf{M}_j - \mathbf{M}_i\|\right)\frac{\mathbf{M}_j - \mathbf{M}_i}{\|\mathbf{M}_j - \mathbf{M}_i\|}}_{\text{repulsion from melanophores}}$$
$$+ \underbrace{\sum_{j=1}^{N_X^d} f^{X^d M}\left(\|\mathbf{X}_j^d - \mathbf{M}_i\|\right)\frac{\mathbf{X}_j^d - \mathbf{M}_i}{\|\mathbf{X}_j^d - \mathbf{M}_i\|}}_{\text{repulsion from dense xanthophores}}$$
$$+ \underbrace{\sum_{j=1}^{N_I^d} f^{I^d M}\left(\|\mathbf{I}_j^d - \mathbf{M}_i\|\right)\frac{\mathbf{I}_j^d - \mathbf{M}_i}{\|\mathbf{I}_j^d - \mathbf{M}_i\|}}_{\text{weak repulsion from dense iridophores}}.$$

Several studies[2,11,41] noted $M$ move away from xanthophores, leading to our $X^d$ repulsion term. Motivated by suggestions in ref.[11], we also specify weak repulsive forces on $M$ due to $I^d$ at short range. Perspectives on the forces $M$ exert on other $M$ differ: experiments[16] showed that $M$ spread out when they have space to move, while Walderich et al.[35] found that $M$ transplanted into *nacre* and albino fish clump together to reach normal densities, even when there is space available. We account for these perspectives by specifying a force that acts between neighboring cells (on the order of 50 μm, the average distance between $M$[16]). This ensures that normal $M$ densities are achieved. Speeds are based on data from ref.[16] suggesting that $M$ migrate roughly 80–100 μm per week.

The ODEs for xanthophore migration, which depends on cell form, are:

$$\frac{d\mathbf{X}_i^d}{dt} = \underbrace{\sum_{j=1, j\neq i}^{N_X^d} f^{X^d X^d}\left(\|\mathbf{X}_j^d - \mathbf{X}_i^d\|\right)\frac{\mathbf{X}_j^d - \mathbf{X}_i^d}{\|\mathbf{X}_j^d - \mathbf{X}_i^d\|}}_{\text{repulsion from dense xanthophores}}$$
$$+ \underbrace{\sum_{j=1}^{N_X^l} f^{X^l X^d}\left(\|\mathbf{X}_j^l - \mathbf{X}_i^d\|\right)\frac{\mathbf{X}_j^l - \mathbf{X}_i^d}{\|\mathbf{X}_j^l - \mathbf{X}_i^d\|}}_{\text{repulsion from loose xanthophores}}$$
$$+ \underbrace{\sum_{j=1}^{N_M} f^{MX^d}\left(\|\mathbf{M}_j - \mathbf{X}_i^d\|\right)\frac{\mathbf{M}_j - \mathbf{X}_i^d}{\|\mathbf{M}_j - \mathbf{X}_i^d\|}}_{\text{repulsion from melanophores}}$$
$$+ \underbrace{\sum_{j=1}^{N_I^d} f^{I^d X^d}\left(\|\mathbf{I}_j^d - \mathbf{X}_i^d\|\right)\frac{\mathbf{I}_j^d - \mathbf{X}_i^d}{\|\mathbf{I}_j^d - \mathbf{X}_i^d\|}}_{\text{attraction to dense iridophores}};$$
$$\frac{d\mathbf{X}_i^l}{dt} = \underbrace{\sum_{j=1, j\neq i}^{N_X^l} f^{X^l X^l}\left(\|\mathbf{X}_j^l - \mathbf{X}_i^l\|\right)\frac{\mathbf{X}_j^l - \mathbf{X}_i^l}{\|\mathbf{X}_j^l - \mathbf{X}_i^l\|}}_{\text{repulsion from loose xanthophores}}$$
$$+ \underbrace{\sum_{j=1}^{N_X^d} f^{X^d X^l}\left(\|\mathbf{X}_j^d - \mathbf{X}_i^l\|\right)\frac{\mathbf{X}_j^d - \mathbf{X}_i^l}{\|\mathbf{X}_j^d - \mathbf{X}_i^l\|}}_{\text{repulsion from dense xanthophores}}.$$

We assume $X^d$ experience a short-range repulsive force from neighboring xanthophores (regardless of cell form) to maintain wild-type cell densities. This agrees with experiments[35] suggesting that xanthophores tend to move into empty space as a cohesive net through differentiation and migration. It should be noted that Walderich et al.[35] describe this behavior as attraction, but we model this as a local repulsive force that accounts for experimentally measured cell-to-cell distances and then decays rapidly to zero. Motivated by suggestions in ref.[11], we also include repulsive forces from $M$ to $X^d$ to account for the clear boundaries between stripe and interstripe regions and attraction by $I^d$. The forces on $X^l$, in contrast, are simply to spread these cells out at appropriate densities on our domain. We prescribe that $X^l$ repel across a slightly longer scale than $X^d$ because $X^l$ appear at a lower density than $X^d$ on the fish skin[15,31].

The ODEs for iridophore movement are:

$$\frac{d\mathbf{I}_i^d}{dt} = \underbrace{\sum_{j=1, j \neq i}^{N_I^d} f^{I^d I^d}\left(\left\|\mathbf{I}_j^d - \mathbf{I}_i^d\right\|\right) \frac{\mathbf{I}_j^d - \mathbf{I}_i^d}{\left\|\mathbf{I}_j^d - \mathbf{I}_i^d\right\|}}_{\text{repulsion from dense iridophores}}$$

$$+ \underbrace{\sum_{j=1}^{N_I^l} f^{I^l I^d}\left(\left\|\mathbf{I}_j^l - \mathbf{I}_i^d\right\|\right) \frac{\mathbf{I}_j^l - \mathbf{I}_i^d}{\left\|\mathbf{I}_j^l - \mathbf{I}_i^d\right\|}}_{\text{repulsion from loose iridophores}};$$

$$\frac{d\mathbf{I}_i^l}{dt} = \underbrace{\sum_{j=1, j \neq i}^{N_I^l} f^{I^l I^l}\left(\left\|\mathbf{I}_j^l - \mathbf{I}_i^l\right\|\right) \frac{\mathbf{I}_j^l - \mathbf{I}_i^l}{\left\|\mathbf{I}_j^l - \mathbf{I}_i^l\right\|}}_{\text{repulsion from loose iridophores}}$$

$$+ \underbrace{\sum_{j=1}^{N_I^d} f^{I^d I^l}\left(\left\|\mathbf{I}_j^d - \mathbf{I}_i^l\right\|\right) \frac{\mathbf{I}_j^d - \mathbf{I}_i^l}{\left\|\mathbf{I}_j^d - \mathbf{I}_i^l\right\|}}_{\text{repulsion from dense iridophores}}.$$

We assume iridophores experience forces only from other iridophores and that these forces are primarily to spread cells out within the iridophore layer on the fish skin. Walderich et al.[35] ablated iridophores in a region of X0 and found that iridophores filled these gaps by migrating horizontally and differentiating. This shows that there is no dorsal–ventral directionality inherent in iridophore interactions; instead, it seems that these cells spread outward from the horizontal myoseptum during development simply because this is where there is space for expansion[9,35].

**Length scales.** Cells interact at both short and long range[19], and our cell birth, death and form-change rules depend on the proportions of cells in short- and long-range neighborhoods. We first describe the short-range scales: let $B_{45}^z$, $B_{75}^z$, $B_{\Delta_{xm}}^z$, and $B_{90}^z$ be the disks of radius 45, 75, $\Delta_{xm} = 82$, and 90 $\mu$m, respectively, centered at $\mathbf{z}$, where $\mathbf{z}$ is the location of the cell of interest and $\Delta_{xm}$ is the average distance between $M$ and $X^d$ at stripe boundaries[16]. Then we introduce the indicator functions:

$1_{B_{45}^z}(\mathbf{A}) = 1$ if the cell at position $\mathbf{A}$ is $\leq 45 \, \mu$m from position $\mathbf{z}$, and 0 otherwise;

$1_{B_{75}^z}(\mathbf{A}) = 1$ if the cell at position $\mathbf{A}$ is $\leq 75 \, \mu$m from $\mathbf{z}$, and 0 otherwise;

$1_{B_{\Delta_{xm}}^z}(\mathbf{A}) = 1$ if the cell at position $\mathbf{A}$ is $\leq \Delta_{xm}$ from $\mathbf{z}$, and 0 otherwise;

$1_{B_{90}^z}(\mathbf{A}) = 1$ if the cell at position $\mathbf{A}$ is $\leq 90 \, \mu$m from $\mathbf{z}$, and 0 otherwise.

In comparison, the distances between neighboring cells reported in ref.[16] vary between 36 and 82 $\mu$m depending on type (see Fig. 2f). We use more than one short length scale for the following reasons: first, because some cells that occupy different layers on the fish skin can appear in the same location on our 2D domain, $B_{45}$ can account for interactions between layers. Second, while measurements of the average distance between two melanophores, between two xanthophores, and between $M$ and $X^d$ are available, we do not know of data on other cell-to-cell distances. We expect that the short-range scales used could be reduced by rescaling the model to account for new data.

Long-range interactions may be regulated by long pseudopodia or airineme extensions[38,39]. Nakamasu et al. showed that some interactions between melanophores and xanthophores occur over at least half a stripe width[19] (adult stripe width is $\approx$600 $\mu$m[38]). We model long-range interactions with an annulus and introduce the indicator function:

$1_{\Omega_{long}^z}(\mathbf{A}) = 1$ if the cell at position $\mathbf{A}$ is between

210 and 250 $\mu$m from position $\mathbf{z}$, and 0 otherwise.

Note airinemes reach up to $\approx$250 $\mu$m[39]. We use a scale under 300 $\mu$m because stripes widen during growth (our observation from images in ref.[36]) and melanophores expand over time[9].

**Melanophore birth.** Recent work has shown that $M$ differentiate from precursors or stem cells, often appearing in the locations of future stripes and exhibiting little movement from there[9,15,35,44,45]. Thus we model $M$ birth as occurring at random locations (e.g., from uniformly distributed precursors, which we do not directly take into account). Rooted in the biological literature[10,19], we specify that $I^d$ and $X^d$ promote $M$ birth at long range and $M$ inhibit it. In particular, each day we select a set number $N_{diff}$ of random locations on the domain, and for each location $\mathbf{z}$, a new $M$ cell appears at $\mathbf{z}$ when there are more $I^d$ and $X^d$ than $M$ cells in a long-range annulus surrounding $\mathbf{z}$ (assuming $\mathbf{z}$ is not locally overcrowded by $M$, $X^d$, $I^d$):

$$\underbrace{\sum_{i=1}^{N_X^d} 1_{\Omega_{long}^z}(\mathbf{X}_i^d) + \sum_{i=1}^{N_I^d} 1_{\Omega_{long}^z}(\mathbf{I}_i^d) > \alpha + \beta \sum_{i=1}^{N_M} 1_{\Omega_{long}^z}(\mathbf{M}_i)}_{\text{long−range differentiation signal}} \text{ and}$$

$$\underbrace{\sum_{i=1}^{N_X^d} 1_{B_{\Delta_{xm}}^z}(\mathbf{X}_i^d) + \sum_{i=1}^{N_M} 1_{B_{\Delta_{xm}}^z}(\mathbf{M}_i) + \sum_{i=1}^{N_I^d} 1_{B_{\Delta_{xm}}^z}(\mathbf{I}_i^d) \leq \eta}_{\text{overcrowding condition}} \Rightarrow \text{melanophore birth at } \mathbf{z},$$

We chose $\beta = 3.5$, the same value used in ref.[24], which was informed by ablation experiments[19]. These ablation experiments[19] showed that xanthophores in interstripes promote $M$ birth and that $M$ at a distance inhibit $M$ birth. We think of $\alpha > 0$ as introducing a time delay between $X^d$ and $I^d$ arrival and $M$ precursor response ($\alpha = 3$). Setting $\alpha = 0$ leads to more frequent breaks in interstripes across many simulations because $M$ immediately differentiate at the central interstripe X0. Since $M$ appear within their own layer on the skin, but only in dark stripes, it is unclear how to account for $M$ overcrowding; thus our overcrowding condition introduces an assumption that $M$ are overcrowded by $I^d$ and $X^d$ (e.g., $X^d$ and $I^d$ inhibit $M$ precursors). Lastly, we place melanophores at 1% of the randomly selected locations when the neighborhood is empty:

$$\sum_{i=1}^{N_M} 1_{B_{\Delta_{xm}}^z}(\mathbf{M}_i) + \sum_{i=1}^{N_X^d} 1_{B_{\Delta_{xm}}^z}(\mathbf{X}_i^d) = \gamma \Rightarrow \text{random melanophore birth at } \mathbf{z},$$

This accounts for *pfeffer;shady* and introduces an additional source of noise into the model.

**Xanthophore and iridophore birth.** Singh et al. showed that iridophores initially differentiate from stem cells to appear at the horizontal myoseptum, but from there they actively divide from existing iridophores[9,14,35]. Xanthophores differentiate from larval xanthophores (loose xanthophores in our model), which are spread across the skin when patterning starts[14,15]. Based on refs.[9,14,15], we model xanthophore and iridophore birth by division of existing cells (when they are not overcrowded). From a modeling perspective, this means xanthophores and iridophores promote their own birth at short range, and each day we simultaneously evaluate every cell on the domain for possible division. If an xanthophore is not overcrowded (if there are not too many $X^d$ and $X^l$ cells in its local neighborhood), we place a new xanthophore next to its parent (randomly in a square of side length 10 $\mu$m centered at the parent cell):

$$\sum_{j=1}^{N_X^d} 1_{B_{\Delta_{xm}}^{X_i^d}}(\mathbf{X}_j^d) + \sum_{j=1}^{N_X^l} 1_{B_{\Delta_{xm}}^{X_i^d}}(\mathbf{X}_j^l) \leq \phi \Rightarrow \text{dense xanthophore appears near parent at } \mathbf{X}_i^d;$$

$$\sum_{j=1}^{N_X^d} 1_{B_{\Delta_{xm}}^{X_i^l}}(\mathbf{X}_j^d) + \sum_{j=1}^{N_X^l} 1_{B_{\Delta_{xm}}^{X_i^l}}(\mathbf{X}_j^l) \leq \psi \Rightarrow \text{loose xanthophore appears near parent at } \mathbf{X}_i^l,$$

where $\phi > \psi$ because xanthophores appear at higher densities in dense form[15], and we assume that loose or dense form is determined by the parent cell. We only consider the presence of other xanthophores when evaluating for overcrowding because xanthophores occur in a separate layer than the other cells. Furthermore, it has been suggested that competition with other xanthophores regulates their number[35]. Because Yamaguchi et al.[4] observed xanthophores appearing at random locations after ablation, we also include a small amount of $X^l$ birth (from precursors) at $N_{rand}$ randomly selected locations when density is very low:

$$\sum_{j=1}^{N_X^d} 1_{B_{xm}^z}(\mathbf{X}_j^d) + \sum_{j=1}^{N_X^l} 1_{B_{xm}^z}(\mathbf{X}_j^l) = \gamma \Rightarrow \text{loose xanthophore born at}$$

randomly selected location $\mathbf{z}$,

where $\gamma = 0$.

Similar to xanthophores, iridophores are born by division of existing iridophores in our model (with the loose or dense form of the daughter cell determined by the parent cell). Each day, we loop through all the iridophores and evaluate if they are overcrowded; if not overcrowded, a new cell of the same form is placed randomly in a square of side length 10 $\mu$m centered at the current iridophore:

$$\sum_{j=1}^{N_I^d} 1_{B_{\Delta_{xm}}^{I_i^d}}(\mathbf{I}_j^d) + \sum_{j=1}^{N_I^l} 1_{B_{\Delta_{xm}}^{I_i^d}}(\mathbf{I}_j^l) \leq \rho \Rightarrow \text{dense iridophore appears near } \mathbf{I}_i^d;$$

$$\sum_{j=1}^{N_I^d} 1_{B_{\Delta_{xm}}^{I_i^l}}(\mathbf{I}_j^d) + \sum_{j=1}^{N_I^l} 1_{B_{\Delta_{xm}}^{I_i^l}}(\mathbf{I}_j^l) \leq \rho \Rightarrow \text{loose iridophore appears near } \mathbf{I}_i^l.$$

Based on the observation that iridophores divide more quickly than xanthophores[15], we choose $\phi, \psi < \rho$.

**Melanophore death.** Melanophores are the only cell type that experiences cell death through competition in our model (xanthophores and iridophores only change form). Motivated by ablation experiments[19] and similar to models[20,24], long-range survival signals from $X^d$ to $M$ and local competition between $X^d$ and $M$ are included. Additionally, during model development, we found it important to include a local survival signal for $M$ in the absence of $X^d$; we chose to model this as a weak support signal from $I^l$ to $M$, but it could also come from other cells in black stripes (for example, Frohnhöfer et al.[11] have hypothesized that L-iridophores, which appear below $M$ and are not included in our model, could be involved in pattern maintenance; see Supplementary Note 1 and Supplementary Fig. 16 for details). Every day, each $\mathbf{M}_i$ is evaluated for possible death due to short-range competition with xanthophores:

$$\sum_{j=1}^{N_X^d} 1_{B_{90}^{M_i}}\left(\mathbf{X}_j^d\right) > \mu \sum_{j=1}^{N_M} 1_{B_{90}^{M_i}}\left(\mathbf{M}_j\right) \Rightarrow \text{death of melanophore at } \mathbf{M}_i,$$

where $\mu = 1.25$. This rule specifies that the melanophore at position $\mathbf{M}_i$ dies when there are too many $X^d$ locally (in the disk of radius 90 μm centered at $\mathbf{M}_i$). We also include the observation[19] that xanthophores in interstripes provide a long-range survival signal to $M$:

$$\sum_{j=1}^{N_M} 1_{\Omega_{\text{long}}^{M_i}}\left(\mathbf{M}_j\right) \geq \xi \sum_{j=1}^{N_X^d} 1_{\Omega_{\text{long}}^{M_i}}\left(\mathbf{X}_j^d\right) \text{ and}$$

$$\sum_{j=1}^{N_I^l} 1_{B_{45}^{M_i}}\left(\mathbf{I}_j^l\right) < \nu \Rightarrow \text{death of cell at } \mathbf{M}_i \text{ with probability } p_{\text{death}} \text{ per day},$$

where $p_{\text{death}} = 0.0333$ based on ref.[19], $\xi = 2$, and $\nu = 3$. This rule prescribes that the melanophore at position $\mathbf{M}_i$ has a chance of dying due to long-range inhibition by $M$ if there are not enough $I^l$ nearby and there are more $M$ than $X^d$ at long range. The inclusion of a local survival signal from $I^l$ to $M$ accounts for *pfeffer*, where $M$ survive despite xanthophores not being present (see Supplementary Note 1).

**Xanthophore form transitions.** Xanthophore transitions between loose and dense became better understood empirically during model development. In particular, $X^l$ were shown to respond to local cues, becoming dense in response to the appearance of $I^d$ and loose in stripes[15,31]. This is related to expression of Csf1 by $I^d$, a factor that promotes xanthophores in dense form[10,28]. With one addition (emphasized below), our model supports these findings. Each day, we simultaneously evaluate every xanthophore for form changing according to:

$$\sum_{j=1}^{N_I^l} 1_{B_{75}^{X_i^d}}\left(\mathbf{I}_j^l\right) > a + \sum_{j=1}^{N_I^d} 1_{B_{45}^{X_i^d}}\left(\mathbf{I}_j^d\right) \Rightarrow \mathbf{X}_i^d \text{ becomes loose,} \tag{1}$$

$$\sum_{j=1}^{N_I^d} 1_{B_{45}^{X_i^l}}\left(\mathbf{I}_j^d\right) + \underbrace{P_i \sum_{j=1}^{N_X^d} 1_{B_{75}^{X_i^l}}\left(\mathbf{X}_j^d\right)}_{\text{additional mechanism}} > b + \sum_{j=1}^{N_I^l} 1_{B_{45}^{X_i^l}}\left(\mathbf{I}_j^l\right) + \sum_{j=1}^{N_M} 1_{B_{75}^{X_i^l}}\left(\mathbf{M}_j\right)$$

$$\Rightarrow \mathbf{X}_i^l \text{ becomes dense,} \tag{2}$$

where $a = 2$, $b = 1$, and $P_i$ is a Bernoulli random variable with mean $p = 0.5$. Rule (1) specifies that $\mathbf{X}_i^d$ transforms to loose when there are more $I^l$ than $I^d$ in its local neighborhood. Rule (2) represents a linear condition: within local neighborhoods, the sum of interstripe cells must be greater than the sum of stripe cells, but $X^d$ cells are factored into the interstripe sum only $p = 50\%$ of the time. These rules were motivated by work[15] suggesting that xanthophores strongly associate with iridophores in dense form and ref.[31] indicating that $M$ also affect xanthophore form locally. Note that images of *nacre;shady* are not experimentally available now, so we do not know what form xanthophores self-select to be in (this is controlled by $p$: if $p = 0$, xanthophores prefer a loose form; the larger $p \in (0, 1]$ is, the more strongly they prefer to be dense). Based on ref.[11], we made the choice to assume *nacre;shady* contains only $X^d$. If *nacre;shady* in fact contains both $X^l$ and $X^d$, then the model is consistent across our derivation set when we set $p = 0$ (see Supplementary Fig. 17 and Supplementary Note 2 for details). We suspect that the additional signal from $X^d$ to $X^l$ in Rule (2) could account for the intermediate xanthophore forms found by Mahalwar et al.[31], published after model development. As additional knowledge about *nacre;shady* and intermediate cell forms emerges, the value of $p$ could be refined further.

Note that our parameters are not meant to suggest, for example, that an xanthophore requires signals from two more loose iridophores than dense iridophores to change form; instead, we think of this as capturing stochasticity on the fish skin (e.g., enough cells must be present for communication to occur between any two). Furthermore, in contrast to our nonlinear iridophore dynamics, which seem to necessarily include long-range interactions to account for the

presence of orange spots in *nacre*, xanthophore form is guided entirely by local communication in our model. This supports experimental suggestions[31] that cell form involves gap junction-dependent interactions.

**Iridophore form transitions.** Each day, we simultaneously evaluate every iridophore on the domain for possible form transitions; conditions for changing between loose and dense depend on the number of cells in local and long-range neighborhoods surrounding the iridophore in question. Because our proposed conditions for iridophore form changes are developed in detail in Results, we only provide their mathematical description here but do not further discuss the rationale behind them. Loose iridophore at $\mathbf{I}_i^l$ may transition to dense under two independent rules. First:

$$\underbrace{\sum_{j=1}^{N_M} 1_{B_{90}^{I_i^l}}\left(\mathbf{M}_j\right) < c}_{[\tilde{A}]} \text{ and } \underbrace{\sum_{j=1}^{N_X^d} 1_{\Omega_{\text{long}}^{I_i^l}}\left(\mathbf{X}_j^d\right) < d}_{[\tilde{B}]} \Rightarrow \mathbf{I}_i^l \text{ transforms to dense,} \tag{3}$$

where we refer to Fig. 5 for the description of rules $[\tilde{A}]$ and $[\tilde{B}]$. $\mathbf{I}_i^l$ may also become dense if it satisfies the condition:

$$\underbrace{\sum_{j=1}^{N_M} 1_{B_{90}^{I_i^l}}\left(\mathbf{M}_j\right) < c}_{[\tilde{A}]} \text{ and } \underbrace{\sum_{j=1}^{N_X^d} 1_{B_{75}^{I_i^l}}\left(\mathbf{X}_j^d\right) > e}_{[\tilde{C}]}, \tag{4}$$

where $c = 3$, $d = 9$, and $e = 3$. Dense iridophore at $\mathbf{I}_i^d$ becomes loose when one (or both) of the two conditions is satisfied. First:

$$\underbrace{\sum_{j=1}^{N_M} 1_{B_{90}^{I_i^d}}\left(\mathbf{M}_j\right) > f}_{[A]} \Rightarrow \mathbf{I}_i^d \text{ transforms to loose,} \tag{5}$$

where $f = c = 3$. Alternatively, $\mathbf{I}_i^d$ can become loose according to:

$$\underbrace{\sum_{j=1}^{N_X^d} 1_{\Omega_{\text{long}}^{I_i^d}}\left(\mathbf{X}_j^d\right) > g}_{[B]} \text{ and } \underbrace{\sum_{j=1}^{N_X^d} 1_{B_{75}^{I_i^d}}\left(\mathbf{X}_j^d\right) < h}_{[C]} \Rightarrow \mathbf{I}_i^d \text{ transforms to loose,} \tag{6}$$

where $g = 5$ and $h = 2$. These parameters were chosen by first identifying the overall form of the rules and rough length scales that best captured high-order features of our derivation set; because our model is constrained to reproduce biological cell densities, these length scales then dictate a certain range of possible parameter values; in an iterative process, we carefully searched this parameter space to target the parameters that produced best agreement across finer and finer details of our derivation set. We expect that these parameters can be rescaled as more becomes known about the timescales over which iridophores interact with other cells and their means of communication (e.g., dendrites). Note that, owing to the form of our conditions, setting $c$, $d$, $e$, $f$, $g$, or $h$ to large values (e.g., larger than the total number of cells) or to negative values (e.g., $-1$ cells), effectively turns off different components of our rules (see Supplementary Methods for the adjustments made to produce Figs. 7 and 8).

**Code availability.** The code we developed to simulate the model in this study is available from the corresponding author (A.V.) on reasonable request. Our code runs in MATLAB 9.3, The MathWorks, Inc., Natick, MA, USA, and all parameters are listed in Supplementary Tables 1–4, Supplementary Methods, and Supplementary Fig. 18.

**Data availability.** No experimental datasets were generated; all simulated results and parameters presented in this study are available from the corresponding author (A.V.) on reasonable request.

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

## Acknowledgements

We thank April Dinwiddie, Anastasia Eskova, Andrey Fadeev, Hans Georg Frohnhöfer, Uwe Irion, Prateek Mahalwar, Ajeet Pratap Singh, and Christiane Nüsslein-Volhard for their invaluable feedback and guidance during model development. A.V. was supported by the NSF through grant no. DMS-1148284, by the NSF Graduate Research Fellowship Program under grant no. DGE-0228243, and by the Mathematical Biosciences Institute and the NSF under grant no. DMS-1440386. B.S. was partially supported by the NSF through grant nos. DMS-0907904 and DMS-1409742.

## Author contributions

A.V. and B.S. developed the model and analyzed results. A.V. carried out simulations and drafted the manuscript. Both authors gave final approval for publication.

## Additional information

**Competing interests:** The authors declare no competing interests.

