## [Peer Review File · Nature Communications]

Reviewers' comments:

Reviewer #1 (Remarks to the Author):

In this work Volkening et al. present a mathematical model for zebrafish stripe formation which accounts for a large set of experimental results, including new findings on the leading role of iridophores in this process. Notably, through stochastic simulations based on this model, they were able to reproduce i) wild type stripe development, ii) the pattern of mutants lacking 1 or 2 pigment cell types, and iii) the dynamics of stripe 'regeneration' after ablation. One of the most important features of this model is the presence of redundant mechanisms underlying the interactions between different pigment cell types. The authors argue convincingly that this redundancy may be at the basis of the developmental robustness of this process.

Another fundamental aspect of this model is the central role attributed to iridophores, a cell type neglected in previous models that mainly focused on the interactions between melanophores and xanthophores. While recent experimental results show that iridophore shape transitions guides the behaviors of the other two pigment cells, the mechanisms controlling these transitions (from dense to loose iridophores and vice versa) are still unknown. The authors propose new potential mechanisms underlying iridophore shape transition and show their efficacy in producing normal stripes through model simulations.

Of striking relevance is also the successful reproduction of other Danio patterns through the modulation of the parameters defining the interaction between iridophores and xanthophores. This astonishing result shows that this model can be used to guide the prediction of other patterns which can be subsequently tested experimentally. Through an in silico screen, the model parameters can be changed until a particular pattern is reached. The combination of parameter modifications can be then used to formulate hypothesis on the biology underlying the mutant pattern under analysis.

This model represents a great advancement and deserves to be published in Nat Comm. While previous models based on Turing reaction-diffusion mechanisms (which ignored the leading role of iridophores and the shape changes of xanthophores and iridophores) failed to explain the developmental robustness of this process, the new model described by Volkening et al. not only account for its robustness against perturbations, but also shows how evolutionary changes may occur through specific modifications of the parameter space of known interactions.

Minor comments:

- The authors use 'early stage mutations' in the manuscript, but this should be changed into 'early stage mutants' (fish carrying the mutation and displaying a mutant phenotype).

- The authors use 'cross'/'crosses' to indicate fish carrying two different mutations, but this should be changed into 'double mutant'/'double mutants'. Please indicate double mutants as in the following example: nacre;pfeffer. nacre x pfeffer is not correct.

-line 45: '...shady, a mutant...' AND NOT '...shady, a mutation...'

-line 62: please write what choker and puma are, as they have not been mentioned before. Perhaps: '..., including the late stage mutants choker and puma, laser ablation experiments, etc...'

-line 102: 'on' instead of 'in'

-Figure 4b': can the authors use 'lacking cell/cells' or similar instead of 'cell birth off'?

- The authors define 'early stage mutants' as mutants lacking one or two cell types (birth and death mutants, nacre, pfeffer, shady). Mutations in these fish are not considered to affect the interactions between pigment cells directly. This definition ('early stage mutants') has so far only been used in Iwashita et al. 2006 (as far as we know). Importantly, these mutants have an early AND late phenotype, whereas "late stage mutants" such as obelix and leopard have no early, but only a late phenotype. The term is used consistently, and perhaps one can let it stand, because we do not have a better term ("mutants lacking cell types" would be more correct, but "cell type mutants", which sounds better, would be again misleading, because mutants with cell autonomy for one cell type would be included).

-Figure 3c. It is difficult to understand what the distance between cells and the length of the green arrow symbolize. Can the authors clarify?

Reviewer #2 (Remarks to the Author):

In this study Volkening and Sandstede extend their prior modeling studies of pigment pattern formation in zebrafish to more explicitly include interactions that may contribute to different iridophore and xanthophore morphologies ("loose" vs. "dense"). The study seeks to integrate a large and growing body of empirical literature into a theoretical framework that can be used to better understand the dynamics of cell behaviors and to make predictions that may guide future experimental approaches. Probably the major contribution of the study is a computational demonstration of pattern robustness owing to redundancies in interactions involving iridophores. I have a few comments for the authors to consider.

1. The authors present some of their results in terms of days post fertilization, primarily for heuristic purposes as they acknowledge the existence of more rigorous staging conventions. Nevertheless, the range of ages used is actually rather late, and more typical of fish growing under suboptimal conditions. What some labs find at 21–75 dpf, others see routinely at 14–28 dpf. To maintain generality and not further confuse an already confusing literature, it would be preferable to simply use named stages or standardized sizes throughout (note that SSL should be reported without mm to distinguish from SL which can differ across strains).

2. On line 88 the authors omit simple diffusion as a possibility. Though recent emphases have been on mechanisms other than diffusion, one cannot exclude the possibility that diffusion or other types of passive movement contribute to interactions, especially for secreted factors like csf1, endothelins or other relatively small molecules.

3. Throughout, the authors refer to mutants by phenotypic names as opposed to their underlying loci. It would be preferable to use the locus names for the sake of consistency in the literature and accessibility to non-specialists. If the mutant names must be used then they should include the several different names that have been employed to minimize confusion with existing literature (e.g, panther/pfeffer, primrose/shady).

4. The statement at line 130, of no interactions being altered in these mutant backgrounds, may not be justifiable without further empirical analysis. I would recommend stating this as a (reasonable) assumption rather than a fact.

5. In the section on evolution, it should be clarified that the *patterns* evolved, not the fish. As written, several statements appear to suggest that loss of function contributed to the species origins, rather than pattern origins within species. Additionally, it should be noted that such early inferences were based on rather simple genetic experiments that could only detect average effects

across alleles and specifics tend to be rather more complicated (e.g., Quigley et al. 2005; Patterson et al. 2014).

6. There is some concern over the use of L and D descriptors for xanthophores in another species like *D. albolineatus*. In fact, the model presented by Patterson et al. 2014 is that a cis-regulatory change in that species drives higher levels of *csf1* expression that, in turn, results in precocious, widespread differentiation of abundant xanthophores that appear at high density across the flank. (The *csf1* experiment in zebrafish was simply a test of whether or not such a change in xanthophore differentiation timing and location could drive a broader pattern change.). Are these abundant and rather dense xanthophores or type, L or D? The interspecies transplants and other analyses in Eom et al. 2015 suggest that xanthophores outside of the residual interstripe in *D. albolineatus* are closer to D than to L. This may be included in the model but, as written, it comes across as ambiguous.

7. Line 327, states that airinemes interact with early larval melanophores. It would be more accurate to state that airinemes interact with both persisting early larval melanophores and newly differentiating adult melanophores, as shown in Eom et al. 2015.

8. Regarding scope and emphasis, the paper really seems to focus on roles for xanthophore and melanophore lineages in driving iridophore morphology. but there should probably be more recognition of roles played by iridophores in driving melanophore and xanthophore morphology, and therefore overall pattern features. For example, Patterson et al. 2013 showed that interstripe iridophores express the *csf1*, which drives differentiation and is likely chemotactic for xanthophores, and misexpression of *csf1* can drive ectopic differentiation of ("D") xanthophores whereas loss of iridophores significantly delays the differentiation of interstripe (D) xanthophores. This suggests an important early role for iridophores in driving pattern, that does not come through as clearly as it might, given the emphasis here on the reciprocal and later interactions.

9. In the supplementary materials the authors cite analyses using an ErbB inhibitor for specific ablation of iridophores. Prior analyses of Budi et al. 2008 and Dooley et al. 2013 show that inhibition of this pathway affects melanophores as well as iridophores. A more specific manipulation of iridophores was obtained by Patterson et al. 2013, who used nitroreductase to ablate the cells (in addition to the analyses of iridophore mutants in that paper and in Frohnhofner et al. 2013).

Reviewer #3 (Remarks to the Author):

A review of "Iridophores as a source of robustness in zebrafish stripes and variability in *Danio* Patterns"

The authors develop and analyze a complex and comprehensive individual-based mathematical model of zebrafish pigmentation patterns. For the first time the authors undertake a comprehensive inclusion of a third type of pigment cells, iridophores, which have been largely overlooked in previous mathematical models of zebrafish pigmentation pattern, and to some extent in the associated biological literature. Since little is known about the role of iridophores in the pattern formation process, the authors generate a number of hypotheses that are consistent with observed patterns (both wild-type stripes and a variety of mutants and mutant crosses). They then test these hypotheses on a different set of mutant patterns/experiments using the same parameter values and find their model to give consistent results.

Further, they alter particular interaction types to produce patterns, which are reminiscent of other species of the *Danio* genus.

The paper represents a significant advance on the current state of modelling of zebrafish

pigmentation patterning and generates experimentally testable hypotheses that lead to genuine biological insight. The impact of the work is likely to be far reaching in both the mathematical modelling and experimental fields.

I have no reservations in recommending this paper for publication.

I do, however, have the following minor comments, which it would be useful for the authors to address before publication.

Line 11 - It would be good to give the names of the two cell types (e.g. Xanthophores and Melanophores) in the abstract for consistency.

Line 16 - replace "is" with "are".

Line 33 but also throughout - "loose shape". Loose refers to the density of the Xanthophores and iridophores and not to the shape of the cells themselves (although they may also have a different shape). Could the authors change this to "loosely configured" or perhaps "sparse" to make the description more accurate?

Line 33 - (yellow or blue, respectively).

Line 34 - (orange or silver, respectively).

Line 45 - Presumably the governing behavior of iridophores is not particularly visible in shady, rather its absence is particularly noticeable?

Line 47 - "... change form normally ..." could the authors be slightly more specific. Are they referring to the change from loose to dense configurations here?

Line 60 and throughout - The authors use the terminology "early stage mutations". I assume the authors mean the phenotype of genetic mutation when looked at an early stage of development? Could the authors clarify or rephrase this as I think the wording is a little confusing.

Line 62 - Should choker and puma be referred to as choker and puma mutants?

Line 65 - "reducing how iridophores react to xanthophores" could the authors be more explicit about what this entails. What is it in the interactions that is reduced?

Lines 81-83 - "To simplify our model, we account for the 3-layer fish skin indirectly by constraining cell birth (to prevent overcrowding) only by cells that would appear in the same layer (M birth is an exception, Fig. 3b)."

Presumably the same is true of any cell movement events which take place? They only feel the short-scale repulsion effects designed to mimic volume exclusion if they are on the same layer?

Why is M birth an exception and how does this relate to figure 3b?

Figure 3 caption - (a) "Notation" is probably not quite the right word for the pictorial representations although I understand what the authors mean.

Figure 3 caption - Is pseudopodia the correct terms for long cellular protrusions or a name for cellular protrusions in general.

Figure 3 caption - The second instance of (e) should be (f).

Line 102 - in vs on

Figure 4 (b) - the authors mention an "expanded central interstripe" in the Nacre mutant, but this is not clear to see in figure 4 (b).

Figure 4 (b) - Similarly conclusion about the uniformity of iridophores for the Pfe-Nac cross seem hard to justify based on the image presented in Fig 4 (b).

Line 130 - The authors state that "no interactions are altered" and that only cells are missing. Is there biological evidence to confirm that no cell-cell interactions are altered, even quantitatively in these mutants?

Line 135 "our derivation set" I think there is a word missing here.

Line 161 "natural form" Is it correct to say the "natural form" here refers to the dense form?

Page 7 - The authors state that the tilde-ed interactions refer to signals transmitted to $I^{\wedge}I$. Could it be that \tilde{B} for example is actually a lack of signal being transmitted to $I^{\wedge}d$?

Figure 6 (a) and (a'). The figures that the authors reproduce are certainly qualitatively similar to the experimental ablation patterns. The formation of a spot of melanophores seems to be quite stark in the experimental figures. Could the authors comment on its significance or otherwise in light of the fact their model does not reproduce it in the examples shown?

Line 294 2D should be 2-D.

Movies - I found the movies very helpful. They would be even more so if they were given names that easy allowed the user to infer what the simulation was showing.

Line 309 - "Biology, however, is after resilience to mutation". This struck me as being out of context with the rest of the paper, but in its informality and its ascribing a purpose of direction to biology.

Lines 333-335 - The authors discuss the possibility of a continuum model. Presumably, this would have to be based on a vastly simplified version of the individual-based model?

Line 340-342 - I don't really understand the sentence about removing the neighborhood and the "cell's velocities normal to these boundaries". Could the authors clarify?

Boundary conditions - Do the boundary conditions make a significant difference to the patterning. Would zero flux boundary conditions be more appropriate than periodic?

Line 356 - I would suggest a word other than varies to describe the growth. Varies makes the distances sound like variation at a single time point.

Line 360 - Where is the origin of growth? Presumably, this matters for non-period BCs but not for periodic?

Below Line 392 Cell=Cells

Page 16 - Remind us what X_0 is as it comes a little out of the blue.

Page 16 - What does it mean biologically for M to be overcrowded by I^d and X^d given that these cells are on different layers.

Page 16 "From a modelling framework" - rephrase?

Page 17 - Which are new Xanthophores placed in a square centered on the parent rather than a circle?

Page 17 - The authors appeal to Frohnhoffer [11] to justify iridophore behavior, but L-iridophores are not explicitly included in the model, so this doesn't make sense to me.

Responses to Reviewer Comments
Alexandria Volkening and Björn Sandstede
May 2018

Thank you for your time and thoughtful comments that have improved our manuscript. In the pages below, we respond to these comments (for each of our 3 reviewers, reviewer comments are copied and italicized). All alterations are highlighted in yellow in the manuscript or supplementary material, and line numbers are given for large changes. Additionally, note that we made the following stylistic changes (not highlighted) to comply with Nat. Commun. guidelines:

- Tables are now written with titles and footnotes.
- Figure panels have been relabelled without apostrophes, and standard deviation bars have been added to Supplementary Fig. 10c-d.
- References are reprinted in the supplementary material (note this means different numbers are assigned to these references in supplementary material).
- Cell positions and other vector-valued variables now appear in bold font.
- We have changed the colors in Fig. 2.
- We have shortened the section headings at lines 186 and 198.
- The initial paragraph summarizing our model has been removed from the Discussion.
- The contents section has been removed from the Supplementary Material.

Reviewer 1:

Remarks to author: *In this work Volkening et al. present a mathematical model for zebrafish stripe formation which accounts for a large set of experimental results, including new findings on the leading role of iridophores in this process. Notably, through stochastic simulations based on this model, they were able to reproduce i) wild type stripe development, ii) the pattern of mutants lacking 1 or 2 pigment cell types, and iii) the dynamics of stripe ‘regeneration’ after ablation. One of the most important features of this model is the presence of redundant mechanisms underlying the interactions between different pigment cell types. The authors argue convincingly that this redundancy may be at the basis of the developmental robustness of this process.*

Another fundamental aspect of this model is the central role attributed to iridophores, a cell type neglected in previous models that mainly focused on the interactions between melanophores and xanthophores. While recent experimental results show that iridophore shape transitions guides the behaviors of the other two pigment cells, the mechanisms controlling these transitions (from dense to loose iridophores and vice versa) are still unknown. The authors propose new potential mechanisms underlying iridophore shape transition and show their efficacy in producing normal stripes through model simulations.

Of striking relevance is also the successful reproduction of other Danio patterns through the modulation of the parameters defining the interaction between iridophores and xanthophores. This astonishing result shows that this model can be used to guide the prediction of other patterns which can be subsequently tested experimentally. Through an in silico screen, the model parameters can be changed until a particular pattern is reached. The combination of parameter modifications can be then used to formulate hypothesis on the biology underlying the mutant pattern under analysis.

This model represents a great advancement and deserves to be published in Nat Comm. While previous models based on Turing reaction-diffusion mechanisms (which ignored the leading role of iridophores and the shape changes of xanthophores and iridophores) failed to explain the developmental robustness of this

process, the new model described by Volkening et al. not only account for its robustness against perturbations, but also shows how evolutionary changes may occur through specific modifications of the parameter space of known interactions.

1. The authors use “early stage mutations” in the manuscript, but this should be changed into “early stage mutants” (fish carrying the mutation and displaying a mutant phenotype).

We have made this change throughout the manuscript and supplementary material wherever we refer to fish carrying a mutation and featuring an altered pattern.

2. The authors use “cross”/“crosses” to indicate fish carrying two different mutations, but this should be changed into “double mutant”/“double mutants”. Please indicate double mutants as in the following example: *nacre;pfeffer*. *nacre x pfeffer* is not correct.

We have made these changes throughout the manuscript.

3. Line 45 should read “shady, a mutant” AND NOT “shady, a mutation”.

We have made this fix.

4. At line 62: please write what *choker* and *puma* are, as they have not been mentioned before. Perhaps write “including the late stage mutants *choker* and *puma*, laser ablation experiments, etc”.

This is a good point, and it was also brought up by Reviewer 3. We have changed line 60 to read “including the *choker* and *puma* mutants”. We decided not to include the term “late-stage” to avoid needing to define the term and because we no longer use “early-stage” (as suggested by your comment 7).

5. At line 102, it should be “on” instead of “in”.

We have corrected this.

6. In Figure 4b’, can the authors use “lacking cell/cells” or similar instead of “cell birth off”?

We agree this is more natural and have replaced “cell birth off” with “cell type(s) lacking” in Fig. 4d (note the original Fig. 4b’ has been relabeled as Fig. 4d to follow Nat. Commun. style requirements).

7. The authors define “early stage mutants” as mutants lacking one or two cell types (birth and death mutants, *nacre*, *pfeffer*, *shady*). Mutations in these fish are not considered to affect the interactions between pigment cells directly. This definition (“early stage mutants”) has so far only been used in Iwashita et al. 2006 (as far as we know). Importantly, these mutants have an early AND late phenotype, whereas “late stage mutants” such as *obelix* and *leopard* have no early, but only a late phenotype. The term is used consistently, and perhaps one can let it stand, because we do not have a better term (“mutants lacking cell types” would be more correct, but “cell type mutants”, which sounds better, would be again misleading, because mutants with cell autonomy for one cell type would be included).

Thank you for this comment. We have followed your suggestion to use “mutants lacking cell types” for *pfeffer*, *nacre*, and *shady* throughout the paper in place of “early-stage mutations”.

8. In Figure 3c, it is difficult to understand what the distance between cells and the length of the green arrow symbolize. Can the authors clarify?

Fig. 3c gives an overview of our differential equations for cell migration: while not to scale, the length of

each green arrow is related to the strength of the force between any two cells. Longer arrows mean stronger forces: for example, we assume melanophores are more strongly repelled from X^d than from I^d , so the arrow for M_i interacting with X_j^d is longer than the arrow for M_i interacting with I_j^d (this is motivated in part by [11], which suggests iridophores have only a minor repulsive effect on melanophores). The distance between cells, also not to scale, is related to the length scale over which cells interact; this highlights that some cells interact over a longer length scale than others (this is motivated by differences in average cell-cell separations; for example, as in Fig. 3f, the average distance between M and X^d is larger than the average distance between 2 melanophores). To address your question, we have added more details to the caption of Fig. 3 (on page 17 of our revised manuscript): “arrow shows how i th cell moves in response to j th cell; length of arrow, while not to scale, indicates force strength (e.g. M are repelled more strongly from xanthophores than iridophores [11]); cell separation symbolizes that some cells are repelled or attracted over longer length scales than others (see Fig. 3f).”

Reviewer 2:

Remarks to author: *In this study Volkening and Sandstede extend their prior modeling studies of pigment pattern formation in zebrafish to more explicitly include interactions that may contribute to different iridophore and xanthophore morphologies (“loose” vs. “dense”). The study seeks to integrate a large and growing body of empirical literature into a theoretical framework that can be used to better understand the dynamics of cell behaviors and to make predictions that may guide future experimental approaches. Probably the major contribution of the study is a computational demonstration of pattern robustness owing to redundancies in interactions involving iridophores. I have a few comments for the authors to consider.*

1. *The authors present some of their results in terms of days post fertilization, primarily for heuristic purposes as they acknowledge the existence of more rigorous staging conventions. Nevertheless, the range of ages used is actually rather late, and more typical of fish growing under suboptimal conditions. What some labs find at 21-75 dpf, others see routinely at 14-28 dpf. To maintain generality and not further confuse an already confusing literature, it would be preferable to simply use named stages or standardized sizes throughout (note that SSL should be reported without mm to distinguish from SL which can differ across strains).*

Thank you for the comment and we understand and sympathize with this point. Because we compare our work to biological results on zebrafish spanning several labs and 15+ years (including some references before SSL was described by Parichy *et al.* in 2009), we believe it is necessary to introduce dpf, SSL, and developmental stages in the manuscript. For our computer code, we needed to make a choice in order to give our model parameters physical meaning: in principle, we could have based our simulation increment in dpf, SSL, or developmental stage; it does not matter for our code, but we had to make a choice. After a base measurement of time increment is chosen, all other measurements are defined in relation to this and can be exactly converted. This is an important difference between *in silico* pattern formation and zebrafish development *in vivo*: while fish age, size, and developmental stage can be somewhat uncoupled *in vivo* and depend on growth rates as you mentioned, these measurements are one and the same in our model. In particular, our dpf can be converted to SSL or developmental stage in the same way inches can be converted to cm. We refer to dpf in some places in the manuscript because we think it is most natural for interdisciplinary readership; for zebrafish specialists, Table 1 provides a means of precisely converting all our measurements in dpf to SSL or developmental stage. To address your comment in the manuscript, we have added additional details to the caption of Table 1 (on page 19 of the revised manuscript); we have also removed mm from our mention of SSL in Table 1 and at line 321 in *Methods*.

2. *On line 88 the authors omit simple diffusion as a possibility. Though recent emphases have been on mechanisms other than diffusion, one cannot exclude the possibility that diffusion or other types of passive movement contribute to interactions, especially for secreted factors like csf1, endothelins or other relatively small molecules.*

We have added diffusion of secreted factors to the list of possible mechanisms underlying cell interactions (with a reference to Patterson *et al.* 2014 [28]); change is at line 86.

3. *Throughout, the authors refer to mutants by phenotypic names as opposed to their underlying loci. It would be preferable to use the locus names for the sake of consistency in the literature and accessibility to non-specialists. If the mutant names must be used then they should include the several different names that have been employed to minimize confusion with existing literature (e.g, panther/pfeffer, primrose/shady).*

We have added more details where we first introduce *nacre*, *pfeffer*, and *shady* at line 120; where we mention *leopard* at line 44; and where we describe *puma* and *choker* at line 209. In particular, we now note that *leopard* encodes connexin 41.8, *pfeffer* encodes csflrA, *nacre* encodes mitfa, *shady* encodes ltk, *puma* encodes bbc3, and *choker* encodes meox1. We have also added references to Lopes *et al.* 2008 and Lister *et al.* 1999. From a mathematical perspective, we admit we are not familiar with locus names, so we have followed the example in Walderich *et al.* 2016 (since this paper appeared in Nat. Commun.) and a supplementary table of mutations in [2]; please let us know if the details we added are not what you had in mind.

4. *The statement at line 130, of no interactions being altered in these mutant backgrounds, may not be justifiable without further empirical analysis. I would recommend stating this as a (reasonable) assumption rather than a fact.*

This is a good point (also related to comment 17 by Reviewer 3). In response to both comments, we have changed this phrase at line 123 to: “Based on transplantation experiments [5, 6, 11], we assume these phenotypes emerge only because cells are missing; crucially, no interactions are altered.”

5. *In the section on evolution, it should be clarified that the “patterns” evolved, not the fish. As written, several statements appear to suggest that loss of function contributed to the species origins, rather than pattern origins within species. Additionally, it should be noted that such early inferences were based on rather simple genetic experiments that could only detect average effects across alleles and specifics tend to be rather more complicated (e.g., Quigley *et al.* 2005; Patterson *et al.* 2014).*

Thank you for these important points. Following your suggestion, we have clarified that we are referring to the evolution of *Danio* patterns in several places: in the caption of Fig. 2 on page 17 of the revised manuscript (“we suggest how the patterns on two other *Danio* fish may have evolved...”); at line 222 (“suggested that zebrafish stripes evolved”); in the section on *Danio* evolution at lines 244 (“Zebrafish (*D. rerio*) stripes are thought to have evolved from other *Danio* fish patterns”) and 248 (“our models suggests the patterns on *D. albolineatus* and *D. margaritatus* could be related to zebrafish...”); in the caption of Fig. 8 (page 17); and in the Discussion at line 301 (“it has been suggested that zebrafish patterns evolved through gain-of-function mutations.”). We have also added a parenthetical note at line 246: “this means that some crosses of zebrafish with other *Danio* fish result in zebrafish-like stripes [8] (note that such early genetic experiments can only detect average effects across alleles)”.

6. *There is some concern over the use of L and D descriptors for xanthophores in another species like *D. albolineatus*. In fact, the model presented by Patterson *et al.* 2014 is that a cis-regulatory change in that species drives higher levels of csf1 expression that, in turn, results in precocious, widespread differentiation*

of abundant xanthophores that appear at high density across the flank. (The *csf1* experiment in zebrafish was simply a test of whether or not such a change in xanthophore differentiation timing and location could drive a broader pattern change.). Are these abundant and rather dense xanthophores or type, L or D? The interspecies transplants and other analyses in Eom *et al.* 2015 suggest that xanthophores outside of the residual interstripe in *D. albolineatus* are closer to D than to L. This may be included in the model but, as written, it comes across as ambiguous.

We appreciate this interesting comment, and it is a thought-provoking point. Before we address it, we first note a terminology difference between our manuscript and Eom *et al.* 2015. We use X^ℓ (loose, L) and X^d (dense, D) to describe xanthophores that appear in wild-type stripes and interstripes, respectively. For the Eom *et al.* study, our X^d correspond to “xanthophores” and our X^ℓ correspond to “xanthoblasts”. As you mentioned, Eom *et al.* highlighted that xanthoblasts (e.g. X^ℓ) extend airinemes toward larval and differentiating adult melanophores, especially between 7-8 SSL. This airineme behavior was shown to be cell-type (or differentiation-state) specific: xanthophores (e.g. X^d) did not typically extend airinemes. Furthermore, cells of the xanthophore lineage in *D. albolineatus* do not frequently extend airinemes [39]; we believe this is the finding you refer to as support that the xanthophores in our *D. albolineatus* pattern should be closer to “D”. Additionally, as you mentioned, Patterson *et al.* 2014 showed xanthophores may differentiate widely across *D. albolineatus* partly because of high *csf1* expression by the fish skin.

From a modeling perspective, the xanthophores in our *D. albolineatus* pattern are type L (X^ℓ) except for near X0. It is important to note, however, that the L and D descriptors in our model refer to agents that behave based on specified rules: for example, an agent of type X^d reacts to other cells based on a different set of rules than an agent of type X^ℓ does. In our current model, there is not a continuum of cell types: if an interaction breaks down (e.g. if we were to remove the rule that X^d are attracted to I^d), we still call these altered agents X^d cells because the remaining model rules they follow are intact. In this sense, if the cells of the xanthophore lineage on *D. albolineatus* behave like X^ℓ with the main exception that they do not produce airinemes, we would consider these X^d agents in our model (as mentioned in the Discussion, our model does not currently include the airineme interactions described by Eom *et al.*).

Thus, from a biological perspective, our predictions about *D. albolineatus* depend on whether the xanthophore-type cells in this fish behave more like the type ‘D’ cells in wild-type zebrafish interstripes or the type ‘L’ cells in stripes. As described in the text, we obtain *D. albolineatus* patterns with primarily type ‘L’ xanthophores and iridophores, if we change only one interaction in our model (remove the long-range signal from interstripe xanthophores for loose iridophores to become dense). If, instead, *D. albolineatus* has iridophores more similar to type ‘L’ but xanthophores more similar to type ‘D’, we would additionally need to change a signal in our model for xanthophores. In particular, we expect we would need to remove the rule that I^ℓ induce X^d to become X^ℓ , and also specify a signal, modeling *csf1* expression by the skin, that X^ℓ across the domain randomly transition to X^d regardless of their local neighborhood. To address your question in the manuscript, we have added a comment in the caption of Fig. 8 (on page 18): “Note we assume *D. albolineatus* contains both X^d and X^ℓ ; if only one type of xanthophore is present (e.g. [28,39] suggest the xanthophores on this fish have some characteristics in common with X^d), we predict *D. albolineatus* pattern evolution involved additional changes to xanthophore form behavior.”

7. Line 327 states that airinemes interact with early larval melanophores. It would be more accurate to state that airinemes interact with both persisting early larval melanophores and newly differentiating adult melanophores, as shown in Eom *et al.* 2015.

We have rephrased this line to read “some early larval and newly differentiating adult melanophores

are known to interact with loose xanthophores through airineme extensions” (line 293 in the revised manuscript).

8. *Regarding scope and emphasis, the paper really seems to focus on roles for xanthophore and melanophore lineages in driving iridophore morphology, but there should probably be more recognition of roles played by iridophores in driving melanophore and xanthophore morphology, and therefore overall pattern features. For example, Patterson et al. 2013 showed that interstripe iridophores express the csf1, which drives differentiation and is likely chemotactic for xanthophores, and misexpression of csf1 can drive ectopic differentiation of (“D”) xanthophores whereas loss of iridophores significantly delays the differentiation of interstripe (D) xanthophores. This suggests an important early role for iridophores in driving pattern, that does not come through as clearly as it might, given the emphasis here on the reciprocal and later interactions.*

Several papers, including the one you mentioned, by the Parichy and Nüsslein-Volhard labs have demonstrated the impact of iridophores on melanophores and xanthophores and begun to uncover the signaling mechanisms (e.g. csf1 expression as you mentioned). However, it seems to us that less is known about the impact of xanthophores and melanophores on iridophores which is why we focused our attention on this issue. Our second reason for focusing the paper on iridophore morphology is space constraints and presentation clarity for an interdisciplinary audience unfamiliar with zebrafish patterning. Your point on how iridophores promote the differentiation of interstripe xanthophores is included in our xanthophore form transition rules, and we mention it at line 374 when we describe these rules. To address your comment in the manuscript, we have added a sentence in the introduction at line 53 to draw the reader’s attention to the scope of the paper: “Because several recent studies [10, 28, 31] have begun to uncover how iridophores signal other cells, we instead focus on helping elucidate the cues iridophores receive.”

9. *In the supplementary materials the authors cite analyses using an ErbB inhibitor for specific ablation of iridophores. Prior analyses of Budi et al. 2008 and Dooley et al. 2013 show that inhibition of this pathway affects melanophores as well as iridophores. A more specific manipulation of iridophores was obtained by Patterson et al. 2013, who used nitroreductase to ablate the cells (in addition to the analyses of iridophore mutants in that paper and in Frohnhofner. et al. 2013).*

We appreciate this helpful comment; we had not realized ErbB also affects melanophores. We show our model results with these experiments by Walderich *et al.* because [35] included ablation timelines that allowed for more direct comparisons with our simulations. In fact, when simulating ablation with a larger iridophore gap in Supplementary Fig. 15d, we found our *M* cells often “wrapped around” the remaining iridophores; this behavior is more akin to Patterson and Parichy [10] than [35] and perhaps could be explained by the additional effects of ErbB on melanophores you mentioned. We have replaced Supplementary Fig. 15c with an image reproduced from Patterson and Parichy [10] and adjusted the figure caption, including added a note that the iridophore ablation [35] in Supplementary Fig. 15a may also impact melanophores. We have also added a citation to [10] whenever we refer to iridophore ablation in the main manuscript.

Reviewer 3:

Remarks to author: *The authors develop and analyze a complex and comprehensive individual-based mathematical model of zebrafish pigmentation patterns. For the first time the authors undertake a comprehensive inclusion of a third type of pigment cells, iridophores, which have been largely overlooked in previous mathematical models of zebrafish pigmentation pattern, and to some extent in the associated biological literature. Since little is known about the role of iridophores in the pattern formation process,*

the authors generate a number of hypotheses that are consistent with observed patterns (both wild-type stripes and a variety of mutants and mutant crosses). They then test these hypotheses on a different set of mutant patterns/experiments using the same parameter values and find their model to give consistent results. Further, they alter particular interaction types to produce patterns, which are reminiscent of other species of the Danio genus. The paper represents a significant advance on the current state of modelling of zebrafish pigmentation patterning and generates experimentally testable hypotheses that lead to genuine biological insight. The impact of the work is likely to be far reaching in both the mathematical modelling and experimental fields. I have no reservations in recommending this paper for publication. I do, however, have the following minor comments, which it would be useful for the authors to address before publication.

1. *It would be good to give the names of the two cell types (e.g. Xanthophores and Melanophores) in the abstract for consistency.*

We have added this to the abstract. To meet the 150 word limit, we have also removed two unnecessary words.

2. *Line 16 - replace “is” with “are”.*

Our model identifies only one set of mechanisms that is consistent with the biological data, so we have replaced “series of mechanisms” with “a set of mechanisms”.

3. *Line 33 but also throughout - “loose shape”. Loose refers to the density of the Xanthophores and iridophores and not to the shape of the cells themselves (although they may also have a different shape). Could the authors change this to “loosely configured” or perhaps “sparse” to make the description more accurate?*

Thank you for this important comment; the editor also raised a related point. The terminology we have seen in the biological literature varies. We received feedback from the Nüsslein-Volhard lab to avoid the term “sparse” (there is a mutant called “sparse” and this would cause confusion). Singh and Nüsslein Volhard 2015 described the form iridophores and xanthophores take as “loose” and “dense” (xanthophore shape, in turn, is described as compact and stellate). “Loose shape” is used to describe iridophores in dark stripes throughout Fadeev *et al.* 2015; however, “loose” also refers to cell density as you mentioned, and it is not fully clear to us whether Fadeev *et al.* meant “shape” as synonymous with “form” or to really mean physical cell extent. For our model, it is not important: we model cells as point masses. Our “shape-transition rules” essentially describe changes in agent type/behavior, not agent shape. In fact, we had originally used the term “form” and only changed it prior to initial submission because we thought “shape” was more intuitive for interdisciplinary readership. In response to your comment, we have gone back to our original terminology and replaced “shape” with “form” throughout the manuscript.

4. *Line 33 - (yellow or blue, respectively).*

We have made this change.

5. *Line 34 - (orange or silver, respectively).*

We have made this change.

6. *Line 45 - Presumably the governing behavior of iridophores is not particularly visible in shady, rather its absence is particularly noticeable?*

Yes, this is true. We have changed this phrase to “The importance of this governing behavior is particularly

noticeable in *shady*” (line 42 in the revised manuscript).

7. Line 47 - “... change form normally ...”: could the authors be slightly more specific. Are they referring to the change from loose to dense configurations here?

We are referring to a change from dense to loose configurations; *leopard* mutant fish feature black spots in an expanded sea of dense iridophores. It seems the dense iridophores at the center of the young fish fail to transition into a loose form and instead trespass into the regions that melanophores occupy in wild-type. In particular, [30] suggests *leopard* may involve a failure of I^d to change into the loose form. Similarly, [29] notes that iridophores in *leopard* fail at taking on a loose form. To address your comment, line 45 now reads “...contains dense iridophores that fail to change into loose form normally”.

8. Line 60 and throughout - The authors use the terminology “early stage mutations”. I assume the authors mean the phenotype of genetic mutation when looked at an early stage of development? Could the authors clarify or rephrase this as I think the wording is a little confusing.

This was also brought up by Reviewer 1, and we agree the terminology was not precise. Following Reviewer 1’s suggestion, we have adopted the term “mutants lacking cell types” in place of “early stage mutants” (or mutations) throughout the paper. This new term clarifies exactly what we mean by *pfeffer*, *nacre*, *shady*, and associated double mutants.

9. Line 62 - Should *choker* and *puma* be referred to as *choker* and *puma* mutants?

This was also brought up by Reviewer 1, and we now refer to “*choker* and *puma* mutants” as suggested.

10. Line 65 - “reducing how iridophores react to xanthophores”: could the authors be more explicit about what this entails. What is it in the interactions that is reduced?

Thank you for this suggestion, and we agree we should be more explicit. We are referring to simplifications in our rules for how xanthophores instruct iridophores to change form. In particular, we find *Danio albolineatus*-like patterns when we remove a long-range signal from dense xanthophores that signals loose iridophores to become dense. We obtain *Danio margaritatus*-like patterns, in turn, when we remove a short-range signal from dense xanthophores that instructs dense iridophores to remain dense. To address your comment and highlight to the reader that we are changing form transition interactions (rather than birth, death or movement rules), we have replaced the line at 63 with “removing specific form-transition signals between xanthophores and iridophores”.

11. Lines 81-83 - “To simplify our model, we account for the 3-layer fish skin indirectly by constraining cell birth (to prevent overcrowding) only by cells that would appear in the same layer (M birth is an exception, Fig. 3b).” Presumably the same is true of any cell movement events which take place? They only feel the short-scale repulsion effects designed to mimic volume exclusion if they are on the same layer? Why is M birth an exception and how does this relate to figure 3b?

The surface layer on the fish skin contains dense and loose xanthophores; the middle layer contains dense and loose iridophores; and the bottom layer contains melanophores. This means that the top layer has cells in both light and dark stripes, as does the middle layer. The bottom layer, however, only has cells in dark stripes. Because we randomly select positions for melanophore birth across the whole domain, if melanophore birth was only constrained by other melanophores, light stripe regions would look empty from a melanophore-perspective, and we would have continued birth of black cells in light regions. The same issue would be present in *pfeffer*, so we prescribed that M birth is constrained by X^d and I^d . In

wild-type, it is possible that competition with X^d would kill many of these melanophores each day after birth, but we chose to bypass this issue in our model rather than have continual turnover of melanophore birth/death. This choice could be interpreted biologically to mean that X^d have an inhibitory effect not only on melanophores but also on their precursors (and that I^d have a local inhibitory effect on M precursors). We note that a similar modeling choice was made in [24]: there we required that M are born where M density is higher than interstripe xanthophore density (short-range activation by M). As for Fig. 3b, you are right, this was a typo; we meant to refer to the cell layers in Fig. 1.

Regarding cell movement: yes, your intuition about short-scale repulsion is correct with one note. As you expect, there are no repulsive forces between interstripe cells in different layers or stripe cells in different layers (e.g. melanophores do not experience forces from loose iridophores or loose xanthophores, since these cells, present in 3 different layers on the fish skin, can occupy the same space in our model). There are, however, three repulsive forces between stripe and interstripe cells in different layers: melanophores and dense xanthophores repel each other, and dense iridophores weakly repel melanophores. This is supported by [2, 11, 41].

To account for your questions, we have made two changes in the manuscript: first, the original quoted phrase at line 81 now reads “...only be cells that would appear in the same layer (see Fig. 1 and note that M birth is an exception)”. Second, we have added two clarifying notes in Methods; (1) in the section on melanophore birth below line 364: “our overcrowding condition introduces an assumption that M are overcrowded by I^d and X^d (e.g. X^d and I^d inhibit M precursors)”. (2) Under the section on cell migration below line 331: “[migration] interactions are only included when necessary for model results and suggested by the biology (for example, there are no repulsive forces between M and I^l because these cells occupy the same region but in different layers on the fish skin).”

12. *Figure 3 caption - (a) “Notation” is probably not quite the right word for the pictorial representations although I understand what the authors mean.*

We have replaced “notation for cell agents” with “symbols used for cell agents”.

13. *Figure 3 caption - Is pseudopodia the correct terms for long cellular protrusions or a name for cellular protrusions in general?*

The term is used to refer to short projections in [23, 41], so you are correct: pseudopodia is not synonymous with long projections. To be more accurate, we have replaced “pseudopodia (long, red)” in the caption of Fig. 3 on page 17 with “long pseudopodia-like projections (red)”. We have also made changes to the two other places we mention pseudopodia: at line 86, we have replaced “longer extensions (pseudopodia or airinemes)” with “longer extensions (long pseudopodia-like projections [38] or airinemes [39]), and in *Methods*, in the Length Scales section below line 361, we have replaced “pseudopodia” with “long pseudopodia”.

14. *Figure 3 caption - The second instance of (e) should be (f).*

Thanks for the catch; we have fixed this.

15. *Line 102 - in vs on*

We have made this change.

16. *Figure 4 (b) - the authors mention an “expanded central interstripe” in the Nacre mutant, but this is*

not clear to see in figure 4 (b). Similarly conclusion about the uniformity of iridophores for the Pfe-Nac cross seem hard to justify based on the image presented in Fig 4 (b).

Note that we have renamed Fig. 4b to 4c to comply with the Nat. Commun. style. It is important to note that mutant phenotypes are very variable [11], so our remarks about these mutants are based both on the empirical descriptions we have read. Frohnhofer *et al.* [11] describe the central interstripe as “prominent” in *nacre*. Importantly, the empirical image of *nacre* in Fig. 4c is a portion of the full fish image reproduced in Supplementary Fig. 6, which much more clearly shows the expanded orange interstripe region: there the central interstripe has expanded to cover the upper half of the fish. To clarify *nacre* in the text, we have added a parenthetical reference at line 142 where we first describe it in the section on proposing iridophore form change mechanisms: “Nacre, in turn, features an expanded central interstripe with rough borders.... (note this can be seen more clearly in Supplementary Fig. 6)”.

For *pfeffer;nacre*, we mean uniformity in the sense that only dense iridophores are present. Unfortunately the lighting in Fig. 4c and Supplementary Fig. 9c may not make this clear. Frohnhofer *et al.* [11] describe a dense layer of iridophores across the fish, but this uniformity is presented most clearly in Fig. 10 of [11], where Frohnhofer *et al.* present a descriptive cartoon of *pfeffer;nacre* containing only silver dense iridophores. Our description of *pfeffer;nacre* at line 143 now reads: “*pfeffer;nacre* has a silver pattern consisting of I^d [11]”.

17. Line 130 - The authors state that “no interactions are altered” and that only cells are missing. Is there biological evidence to confirm that no cell-cell interactions are altered, even quantitatively in these mutants?

This is also related to point 4 by Reviewer 2, who suggested we should soften this statement to be a reasonable assumption. The biological support comes in the form of transplantation experiments: introducing the missing cell type into *nacre*, *pfeffer*, or *shady* leads to partial recovery of stripe patterns [5, 6, 11]. This suggests *nacre*, *pfeffer*, and *shady* are cell-autonomous to xanthophores, melanophores, and iridophores, respectively [11]. In response to your comment, following Reviewer 2’s suggestion, we have changed this line at 123 to “Based on transplantation experiments [5, 6, 11], we assume these phenotypes emerge only because cells are missing; crucially, no interactions are altered.”

18. Line 135 - “our derivation set” I think there is a word missing here.

We have taken out this phrase and line 127 now reads “[we] search for a single model that can simultaneously explain the development of these altered patterns and wild-type stripes”.

19. Line 161 - “natural form”: Is it correct to say the “natural form” here refers to the dense form?

Yes, we were referring to “dense” as the natural form of iridophores; this is because, when no other cell types are present, iridophores in *nacre;pfeffer* appear only in the dense form. This suggests that, in the absence of other cell signals, iridophores prefer to be dense. To be more precise, we have replaced “natural form” with “dense form”.

20. Page 7 - The authors state that the tilde-ed interactions refer to signals transmitted to I^ℓ . Could it be that \tilde{B} for example is actually a lack of signal being transmitted to I^d ?

Absolutely, and this is one interpretation we have in mind for this model rule. In particular, M transmit a signal to I^ℓ to remain loose, and when this signal is missing, I^ℓ become dense. We still consider this a “signal transmitted to” I^ℓ in some sense, because I^ℓ are the agents acting in response to the cues (or lack

of cues) from their surrounding neighborhood. To account for this in the text, we have added a note in the caption of Fig. 5 on page 17: “note that [A], [B], and $[\tilde{C}]$ may be active signals, while [C], $[\tilde{A}]$, and $[\tilde{B}]$ could be interpreted as passive cues that represent the absence of a signal from the crossed-out cell type.”

21. *Figure 6 (a) and (a'). The figures that the authors reproduce are certainly qualitatively similar to the experimental ablation patterns. The formation of a spot of melanophores seems to be quite stark in the experimental figures. Could the authors comment on its significance or otherwise in light of the fact their model does not reproduce it in the examples shown?*

Yamaguchi *et al.* [4] conducted two ablation experiments, one of which we reproduce in Fig. 6 and the other we reproduce in Supplementary Fig. 14. These were early experiments that helped show zebrafish patterns form autonomously and do not fill in some sort of pre-pattern. Importantly, the spot on Fig. 6a is not discussed by Yamaguchi *et al.* - they highlight that the pattern width is maintained, though the directionality is lost. Moreover, while Yamaguchi *et al.* [4] offer a single example of their first type of ablation experiment (reproduced in Fig. 6a), they provide a few movies of different versions of their second ablation experiment (Supplementary Fig. 14). These videos show a lot more variability; in one, we see a black bell curve stripe form, and, in another, a lower black stripe connects with an upper black stripe. Thus, we believe the light spot in Fig. 6a should not be heavily focused on, as this is the result of a single experiment, and the second ablation experiment demonstrates the amount of variability present. We have rewritten our statement in the caption of Fig. 6a on page 18 to better highlight the key characteristic of these ablation patterns: “Ablation of a rectangular region *in vivo* [4] and (b) ablation *in silico* (also see Supplementary Movie 7). In both cases, the resulting patterns are characterized by stripes with normal width but lost directionality.”

22. *Line 294: 2D should be 2-D.*

We have removed this paragraph, which summarized our model, from the Discussion to avoid repeating material from Results (a Nat. Commun. style requirement).

23. *Movies - I found the movies very helpful. They would be even more so if they were given names that easy allowed the user to infer what the simulation was showing.*

Unfortunately the movie names are not something we can change based on Nat. Commun. guidelines for supplementary material (our understanding is that all supplementary items must be numbered and referred to by type). If published, we expect the movies will be presented online alongside their captions, and this should clarify what each simulation is showing.

24. *Line 309 - “Biology, however, is after resilience to mutation”. This struck me as being out of context with the rest of the paper, but in its informality and its ascribing a purpose of direction to biology.*

We have changed this line (now at line 276) to read: “our mathematical intuition is to search for a single elegant rule. From a biological perspective, however, resilience to mutation is key”.

25. *Lines 333-335 - The authors discuss the possibility of a continuum model. Presumably, this would have to be based on a vastly simplified version of the individual-based model?*

Yes, the continuum model would be based on a simplified version of our agent-based model. The PDEs we have in mind would involve reaction terms (modeled using convolutions for long-range interactions) based on our agent-based rules.

26. *Line 340-342 - I don't really understand the sentence about removing the neighborhood and the "cell's velocities normal to these boundaries". Could the authors clarify? Boundary conditions - Do the boundary conditions make a significant different to the patterning. Would zero flux boundary conditions be more appropriate than periodic?*

Our boundary conditions at the top and bottom of the domain are essentially zero flux. When a cell is within $50\ \mu\text{m}$ of the top or bottom boundary, we remove the vertical component of its velocity if it is pointing in the direction outside of the domain. For example, suppose a cell is located at point $(x, y) = (100, 30)$ and the forces it experiences from other cells lead it to have a velocity $\mathbf{v} = (10, -5)$. The lower boundary of our domain is at $y = 0$, so this cell is within $50\ \mu\text{m}$ of the boundary and we will adjust its velocity to $\mathbf{v} = (10, 0)$. If, on the other hand, its original velocity at the same point was $\mathbf{v} = (10, 5)$, we would make no change.

We expect the choice of boundary conditions makes very little impact on our model results in terms of cell movement. However, the periodic boundary conditions are helpful in our rules for cell birth, death, and form changes: for example, the long-range annulus and short-range disk used in our model rules extend outside of the domain for cells near the boundary. If our boundary conditions were not periodic, this would cause cells located near the edge of the domain to be exposed to different cell densities than elsewhere. We do not think periodic boundary conditions are less appropriate than zero flux for this reason, particularly because we are concerned with pattern formation at the center of the fish. In the future, it would be interesting to consider how patterns form in other regions (for example, near the eye or where the body meets the tailfin) where boundary conditions may be more critical biologically.

In response to your comments, our description of boundary conditions in *Methods* at line 305 now reads: "We use periodic boundary conditions lengthwise for cell movement, birth, death, and form changes. Our boundary conditions at the top and bottom of the domain are no flux: we implement this by removing the vertical component of a cell's velocity if the cell is within $50\ \mu\text{m}$ of the top (respectively, bottom) edge of the domain and the y-coordinate of its velocity points upward (respectively, downward)."

28. *Line 356 - I would suggest a word other than varies to describe the growth. Varies makes the distances sound like variation at a single time point.*

We have replaced "varies" with "increases".

29. *Line 360 - Where is the origin of growth? Presumably, this matters for non-period BCs but not for periodic?*

We model growth as stretching the distances between cells (e.g. based on the assumption that skin cells in zebrafish differentiate everywhere across the fish). Regarding the details, our cell positions have x-coordinates falling between 0 and the current domain length $L(t)$. Similarly, the y-coordinates of our cells fall between 0 and the current domain height $H(t)$. If the domain grows by ΔL length-wise and ΔH vertically in one day, the cell at position (x, y) is updated as

$$x_{t+1} = x_t \times \frac{L(t) + \Delta L}{L(t)} \qquad y_{t+1} = y_t \times \frac{H(t) + \Delta H}{H(t)}.$$

Thus, there is not really an origin of growth in either direction: the distances between the x-coordinates of all cells are scaled uniformly. For vertical growth, again, the distances between the y-coordinates of all cells increase uniformly based on the original distance between the cells. To address your comment, we

have added a note at line 326 that “cell positions are stretched across the domain (this is done by multiplying a cell’s x - and y -coordinates by respective scaling factors).”

30. Below Line 392 Cell=Cells

Thank you for catching this typo; we have fixed it.

31. Page 16 - Remind us what X^d is as it comes a little out of the blue.

We have changed this to read “the central interstripe X^d ”.

32. Page 16 - What does it mean biologically for M to be overcrowded by I^d and X^d given that these cells are on different layers.

This comment is related to our response to your earlier question 11. As mentioned there, this could be interpreted biologically to mean that X^d and I^d prevent melanophore precursors from differentiating; it essentially serves as a local inhibitory effect on M by interstripe cells (local inhibition of M by interstripe xanthophores was also included in [24]). To highlight this, we have added a note in *Methods* below line 364 in the section on melanophore birth: “our overcrowding condition introduces an assumption that M are overcrowded by I^d and X^d (e.g. X^d and I^d inhibit M precursors)” .

33. Page 16 “From a modelling framework” - rephrase?

We have changed this to “From a modeling perspective”.

34. Page 17 - Which are new Xanthophores placed in a square centered on the parent rather than a circle?

We initially used a square when coding up the model and this choice just remained throughout our work. In practice, though, it does not matter if a circle or square is used for our model. This is because the width of the square we used is small (10 μm) compared to the radius over which xanthophores experience forces from other xanthophores. In our model, all cell birth, death, and form change behaviors occur simultaneously, and then cell movement is simulated. Thus, each new xanthophore is placed in a square region around its parent, but migration forces immediately move it away from its parent before it interacts with other cells. The difference in movement for xanthophores placed in a square or circle region would be very small, so in practice the use of a square or circle for cell birth does not make a difference.

35. Page 17 - The authors appeal to Frohnhofner [11] to justify iridophore behavior, but L-iridophores are not explicitly included in the model, so this doesn’t make sense to me.

L-iridophores, located in black stripes, become a part of zebrafish patterns only after the basic stripe pattern has formed, and it has been hypothesized that they may be involved in maintenance [11]. In our model development process, we found it was necessary to include some sort of local survival signal to keep melanophores alive when xanthophores are absent, so that the pattern does not slowly degrade (as discussed in Supplementary Note 1). This local survival signal could take many forms; it simply needs to operate in a nearby neighborhood of melanophores regardless of whether xanthophores are present or absent. On the fish skin, this survival signal could come from M , I^ℓ , or L-iridophores - anything but loose xanthophores in the nearby vicinity of M . We chose to prescribe it to I^ℓ , but it is possible this is an indirect way of capturing signals transmitted by the L-iridophores that would be beneath them on the fish skin.

To address your comment, we have made 2 changes: (1) In *Methods*, below line 369 on page 11, the text you mentioned now reads “we found it important to include a local survival signal for M in the absence of

X^d ; we chose to model this as a weak support signal from I^ℓ to M , but it could also come from other cells in black stripes (for example, Frohnhöfer *et al.* [11] have hypothesized that L-iridophores, which appear below M and are not included in our model, could be involved in pattern maintenance;”). (2) We have also added a clarification regarding this local melanophore survival signal in Supplementary Note 1.

REVIEWERS' COMMENTS:

Reviewer #1 (Remarks to the Author):

I have checked the revisions of the Nature Communications manuscript NCOMMS-17-33592A and approve it. All my suggestions and requests have been followed. In my opinion, the manuscript is ready for publication.

Reviewer #2 (Remarks to the Author):

The authors have adequately addressed by comments.

Reviewer #3 (Remarks to the Author):

Thanks for your thorough response to my questions. I am happy that you have addressed all the concerns I had.